# Realistic in silico generation and augmentation of single-cell RNA-seq data using generative adversarial networks

Matteo Marouf[1,5], Pierre Machart [1,5], Vikas Bansal [1], Christoph Kilian [1,2], Daniel S. Magruder[1,3], Christian F. Krebs[2] & Stefan Bonn [1,4]*

A fundamental problem in biomedical research is the low number of observations available, mostly due to a lack of available biosamples, prohibitive costs, or ethical reasons. Augmenting few real observations with generated in silico samples could lead to more robust analysis results and a higher reproducibility rate. Here, we propose the use of conditional single-cell generative adversarial neural networks (cscGAN) for the realistic generation of single-cell RNA-seq data. cscGAN learns non-linear gene–gene dependencies from complex, multiple cell type samples and uses this information to generate realistic cells of defined types. Augmenting sparse cell populations with cscGAN generated cells improves downstream analyses such as the detection of marker genes, the robustness and reliability of classifiers, the assessment of novel analysis algorithms, and might reduce the number of animal experiments and costs in consequence. cscGAN outperforms existing methods for single-cell RNA-seq data generation in quality and hold great promise for the realistic generation and augmentation of other biomedical data types.

[1] Institute of Medical Systems Biology, University Medical Center Hamburg-Eppendorf, Hamburg, Germany. [2] Center for Internal Medicine, III. Medical Clinic and Polyclinic, University Medical Center Hamburg-Eppendorf, Hamburg, Germany. [3] Genevention GmbH, Goettingen, Germany. [4] German Center for Neurodegenerative Diseases, Tuebingen, Germany. [5]These authors contributed equally: Matteo Marouf, Pierre Machart. *email: sbonn@uke.de

Biological systems are usually highly complex, as intracellular and intercellular communication, for example, are orchestrated via the non-linear interplay of tens to hundreds of thousands of different molecules[1]. Recent technical advances have enabled scientists to scrutinize these complex interactions, measuring the expression of thousands of genes at the same time, for instance[2]. Unfortunately, this complexity often becomes a major hurdle as the number of observations can be relatively small, due to economical or ethical considerations or simply because the number of available patient samples is low. Next to technically induced measurement biases, this problem of too few observations, in the face of many parameters, might be one of the most prominent bottlenecks in biomedical research[1]. Thus, a small sample size might not reflect the population well, an imbalance that can decrease the reproducibility of experimental results[3].

While the number of biological samples might be limited, realistic in silico generation of observations could accommodate for this unfavorable situation. In practice, in silico generation has seen success in computer vision when used for data augmentation, whereby in silico-generated samples are used alongside the original ones to artificially increase the number of observations[4]. In this manuscript, we focus on augmenting real with newly generated samples, in their original high-dimensional gene space, and whose distribution mimics the original data distribution. While classically, data modeling relies on a thorough understanding of the priors on invariants underlying the production of such data, current methods of choice for photorealistic image generation rely on deep learning-based generative adversarial networks (GANs)[5–8] and variational autoencoders (VAEs)[9,10].

GANs involve a generator that outputs realistic in silico-generated samples. This is achieved with a neural network that learns to transform a simple, low-dimensional distribution into a high-dimensional distribution that is virtually indistinguishable from the real training distribution (Supplementary Fig. 1).

While data augmentation has been a recent success story in various fields of computer science, the development and usage of GANs and VAEs for omics data augmentation has yet to be investigated. As a proof of concept that realistic in silico generation could potentially be applied to biomedical omics data, we focus on the generation of single-cell RNA (scRNA) sequencing data using GANs. scRNA sequencing has made it possible to evaluate genome-wide gene expression of thousands to millions of cells in a single experiment[11]. This detailed information across genes and cells opens the door to a much deeper understanding of cell type heterogeneity in a tissue, cell differentiation, and cell type-specific disease etiology.

In this manuscript, we establish how a single-cell GAN (scGAN) can be leveraged to generate realistic scRNA-seq data. We further demonstrate that our scGAN can use conditioning (cscGAN) to produce specific cell types or subpopulations, on-demand. Finally, we show how our models can successfully augment sparse cell populations to improve the quality and robustness of downstream classification. To the best of our knowledge, this constitutes the first attempt to apply these groundbreaking methods for the augmentation of sequencing data.

## Results

### Realistic generation of scRNA-seq data using an scGAN.
Given the great success of GANs in producing photorealistic images, we hypothesize that similar approaches could be used to generate realistic scRNA-seq data (i.e. matrices where each row corresponds to a cell and each column to the expression level of a gene). In this work "realistic" is referring to the generation of data that mimics the distribution of the real data, in their original space, without merely replicating them. To distinguish experimental scRNA-seq data from data produced by GANs we will use the terms "real" and "generated" cells, respectively.

To build and evaluate different GAN models for scRNA-seq data generation we used a peripheral blood mononuclear cell (PBMC) scRNA-seq dataset with 68,579 cells[12] (Supplementary Table 1, Methods). The PBMC dataset contains many distinct immune cell types, which yield clear clusters that can be assigned their cell type identity with marker genes (genes specifically expressed in a cluster). The aforementioned features of the PBMC dataset make it ideal for the evaluation of our scGAN performance.

Since it is notoriously difficult to evaluate the quality of generative models[13,14] we used four evaluation criteria inspired by single-cell data analysis: t-SNE, marker gene correlation, maximum mean discrepancy (MMD), and classification performance (see Methods for evaluation details). These metrics are used as quantitative and qualitative measures to assess the synthesized cells. Based on these criteria, the best performing single-cell GAN (scGAN) model was a GAN minimizing the Wasserstein distance[15], relying on two fully connected neural networks with batch normalization (Supplementary Fig. 1). We found that the quality of the generated cells greatly improved when the training cells were scaled to exhibit a constant total count of 20,000 reads per cell. In addition to this preprocessing step, we added a custom library-size normalization (LSN) function to our scGAN's generator so that it explicitly outputs generated cells with a total read count equal to that of the training data (20,000 reads per cell) (Supplementary Fig. 1). Our LSN function greatly improved training speed and stability and gave rise to the best performing models based on the aforementioned metrics. Further details of the model selection and (hyper)-parameter optimization can be found in the Methods section.

For a qualitative assessment of the results, we used t-SNE[16,17] to obtain a two-dimensional representation of generated and real cells from the test set (Fig. 1a–c, Supplementary Fig. 2). The scGAN generates cells that represent every cluster of the data it was trained on and the expression patterns of marker genes are accurately learned by scGAN (Supplementary Fig. 3).

Furthermore, the scGAN is able to model intergene dependencies and correlations, which are a hallmark of biological gene-regulatory networks[18]. To prove this point we computed the correlation and distribution of the counts of cluster-specific marker genes (Fig. 1d) and 100 highly variable genes between generated and real cells (Supplementary Fig. 4). We then used SCENIC[19] to understand if scGAN learns regulons, the functional units of gene-regulatory networks consisting of a transcription factor (TF) and its downstream regulated genes. scGAN trained on all cell clusters of the Zeisel dataset[20] (see Methods) faithfully represent regulons of real test cells, as exemplified for the Dlx1 regulon in Supplementary Fig. 4G–J, suggesting that the scGAN learns dependencies between genes beyond pairwise correlations.

To show that the scGAN generates realistic cells, we trained a Random Forest (RF) classifier[21] to distinguish between real and generated data. The hypothesis is that a classifier should have a (close to) chance-level performance when the generated and real data are highly similar. Indeed the RF classifier only reaches 0.65 area under the curve (AUC) when discriminating between the real cells and the scGAN-generated data (blue curve in Fig. 1e) and 0.52 AUC when tasked to distinguish real from real data (positive control).

Finally, we compared the results of our scGAN model to two state-of-the-art scRNA-seq simulations tools, Splatter[22] and SUGAR[23] (see Methods for details). While Splatter models some marginal distribution of the read counts well (Supplementary Fig. 5), it struggles to learn the joint distribution of these counts,

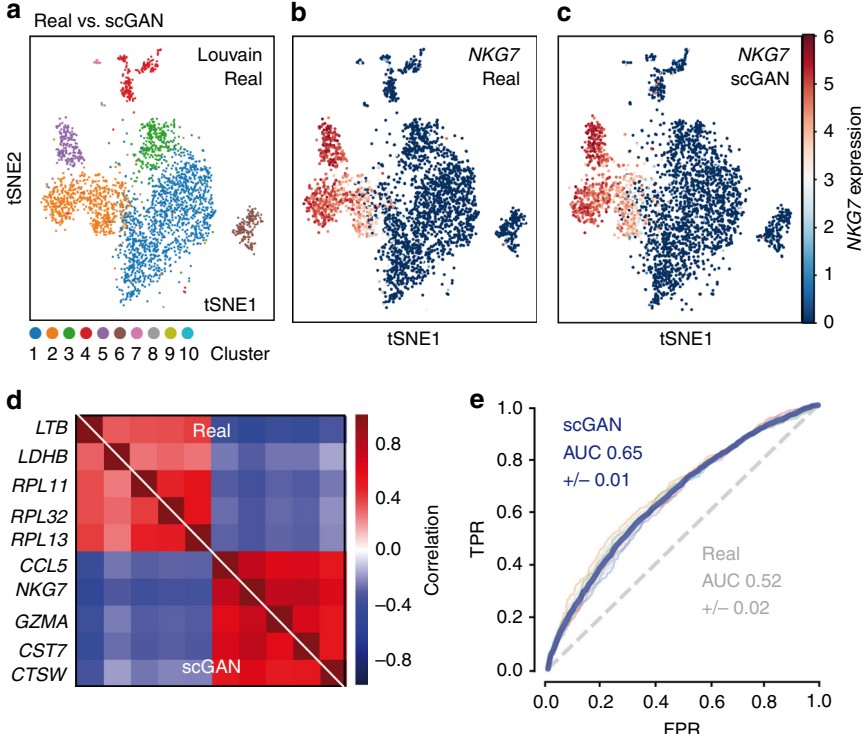

**Fig. 1 Evaluation of the scGAN-generated PBMC cells. a–c** t-SNE visualization of the Louvain-clustered real cells (**a**) and the *NKG7* gene expression in real (**b**) and scGAN-generated (**c**) cells. **d** Pearson correlation of marker genes for the scGAN-generated (bottom left) and the real (upper right) data. **e** Cross-validation ROC curve (true positive rate against false positive rate) of an RF classifying real and generated cells (scGAN in blue, chance-level in gray).

as observed in t-SNE visualizations with one homogeneous cluster instead of the different subpopulations of cells of the real data, a lack of cluster-specific gene dependencies, and a high MMD score (129.52) (Supplementary Table 2, Supplementary Fig. 4). SUGAR, on the other hand, generates cells that overlap with every cluster of the data it was trained on in t-SNE visualizations and accurately reflects cluster-specific gene dependencies (Supplementary Fig. 6). SUGAR's MMD (59.45) and AUC (0.98), however, are significantly higher than the MMD (0.87) and AUC (0.65) of the scGAN and the MMD (0.03) and AUC (0.52) of the real data (Supplementary Table 2, Supplementary Fig. 6). It is worth noting that SUGAR can be used, like here, to generate cells that reflect the original distribution of the data. It was, however, originally designed and optimized to specifically sample cells belonging to regions of the original dataset that have a low density, which is a different task than what is covered by this manuscript. While SUGAR's performance might improve with the adaptive noise covariance estimation, the runtime and memory consumption for this estimation proved to be prohibitive (see Supplementary Fig. 6F–I and Methods).

The results from the t-SNE visualization, marker gene correlation, MMD, and classification corroborate that the scGAN generates realistic data from complex distributions, outperforming existing methods for in silico scRNA-seq data generation. The realistic modeling of scRNA-seq data entails that our scGAN does not denoise nor impute gene expression information, while they potentially could[24]. Nevertheless, an scGAN that has been trained on imputed data using MAGIC[25] generates realistic imputed scRNA-seq data (Supplementary Fig. 7). Of note, the fidelity with which the scGAN models scRNA-seq data seems to be stable across several tested dimensionality reduction algorithms (Supplementary Fig. 8).

**Realistic modeling across tissues, organisms, and data size**. We next wanted to assess how faithful the scGAN learns very large, more complex data of different tissues and organisms. We therefore trained the scGAN on the currently largest published scRNA-seq dataset consisting of 1.3 million mouse brain cells and measured both the time and performance of the model with respect to the number of cells used (Supplementary Table 1, Supplementary Fig. 9). Qualitative assessment using t-SNE visualization shows that the scGAN generates cells that represent every cluster of the data it was trained on. The expression patterns of marker genes are accurately learned (Supplementary Fig. 9).

The actual time required to train an scGAN depends on the data size and complexity and on the computer architecture used, necessitating at least one high-performance GPU card. However, it should be noted that scGAN uses batch training so that its memory consumption does not depend on the number of cells and its runtime scales linearly, at worst, with it.

Our results demonstrate that the scGAN performs consistently well on scRNA-seq datasets from different organisms, tissues, and with varying complexity and size, learning realistic representations of millions of cells.

**Conditional generation of specific cell types**. scRNA-seq in silico data generation reaches its full potential when specific cells of interest could be generated on demand, which is not directly possible with the scGAN model. This conditional generation of cell types could be used to increase the number of a sparse, specific population of cells that might represent only a small fraction of the total cells sequenced.

While specific cell types of interest can be obtained by scGAN cell generation followed by clustering and cell selection, we developed and evaluated various conditional scGAN (cscGAN)

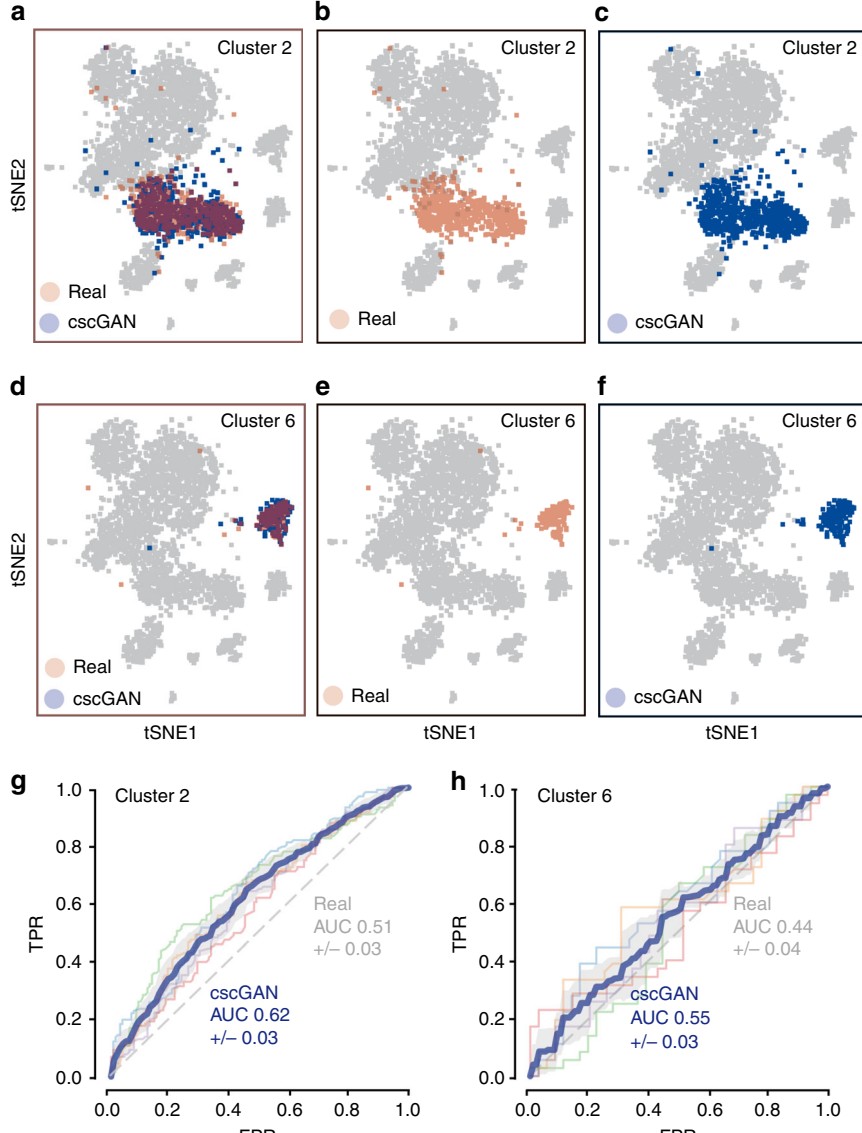

**Fig. 2 Evaluation of the conditional generation of PBMC cells. a–c** t-SNE visualization of cluster 2 real cells (red, panels **a**, **b**), cluster 2 generated cells (blue, panels **a**, **c**), and other real cells (gray, all panels). **d–f** Same as **a–c** for cluster 6 cells. **g** Cross-validation ROC curve of an RF classifying cluster 2 real from cscGAN-generated cells (cscGAN in blue, chance-level in gray). **h** Same as **g** for cluster 6 cells.

architectures that can directly generate cell types of interest. Common to all these models is that the cscGAN learns to generate cells of specific types while being trained on the complete multiple cell type dataset. The cell type information is then associated to the genes' expression values of each cell during the training. These tags can then be used to generate scRNA-seq data of a specific type, in our case of a specific cluster of PBMCs. The best performing cscGAN model utilized a projection discriminator[26], along with Conditional Batch Normalization[27] and an LSN function in the generator. Again, model architecture selection and optimization details can be found in the Methods.

Model performance was assessed on the PBMC dataset using t-SNE, marker gene correlation, and classification. The cscGAN learns the complete distribution of clusters of the PBMC data (Fig. 2) and can conditionally represent each of the ten clusters on demand. The t-SNE results for the conditional generation of cluster 2 and cluster 6 cells are shown in Fig. 2a–c and Fig. 2d–f, respectively. Figure 2a–c highlights the real (red, panels a and b) and generated (blue, panels a and c) cells for cluster 2, while the

real cells of all other clusters are shown in gray. The cscGAN generates cells that are overlapping with the real cluster of interest in the t-SNE visualizations. In addition, the cscGAN also accurately captures inter- and intra-cluster gene–gene dependencies as visualized in the marker gene correlation plots in Supplementary Fig. 10. The assumption that the cscGAN generates conditional cells that are very similar to the real cells of the cluster of interest is substantiated in the final classification task. An RF classifier reaches an AUC between 0.62 (cluster 2, Fig. 2g) and 0.55 (cluster 6, Fig. 2h) when trying to distinguish cluster-specific cscGAN-generated cells from real cells, a value that is reasonably close to the perfect situation of random classification (AUC of 0.5) (Supplementary Table 4). The MMD distances between cscGAN generated and real cells is 0.286 (cluster 2, Fig. 2g) and 0.238 (cluster 6, Fig. 2h) while distances between real and real cells (positive control) were 0.037 and 0.129, respectively (Supplementary Table 3).

It is interesting to observe that the cscGAN and scGAN generate cells of very similar quality, as an RF classifier reaches an AUC of 0.61 (MMD of 0.674) to distinguish between

cscGAN-generated and real cells and an AUC of 0.65 (MMD of 0.547) for scGAN-generated and real cells (differences not significant).

The results of this section demonstrate that the cscGAN can generate high-quality scRNA-seq data for specific clusters or cell types of interest, while rivaling the overall representational power of the scGAN. Importantly, the fidelity with which the cscGAN models scRNA-seq data seems to be independent of the tested Louvain and K-means clustering algorithms (Supplementary Fig. 11, Supplementary Table 5).

**Improved classification of sparse cells using augmented data**. We now investigate how we can use the conditional generation of cells to improve the quality and robustness of downstream classification of rare cell populations. The underlying hypotheses are two-fold. (i) A few cells of a specific cluster might not represent the cell population of that cluster well, potentially degrading the quality and robustness of downstream classification. (ii) This degradation might be mitigated by augmenting the rare population with cells generated by the cscGAN. The base assumption is that the cscGAN might be able to learn good representations for small clusters by using gene expression and correlation information from the whole dataset.

To test the two parts of our hypothesis, we first artificially reduce the number of cells of the PBMC cluster 2 (downsampling) and observe how it affects the ability of an RF model to accurately distinguish cells from cluster 2 from cells of other clusters. In addition, we train the cscGANs on the same downsampled datasets, generate cells from cluster 2 to augment the downsampled population, retrain an RF with this augmented dataset, and measure the gain in their ability to correctly classify the different populations.

More specifically, cluster 2 comprises 15,008 cells and constitutes the second largest population in the PBMC dataset. Such a large number of cells makes it possible to obtain statistically sound classification results. By deliberately holding out large portions of this population, we can basically quantify how the results would be affected if that population was arbitrarily small. We produce eight alternate versions of the PBMC dataset, obtained by downsampling the cluster 2 population (keeping 50%, 25%, 10%, 5%, 3%, 2%, 1%, and 0.5% of the initial population) (Supplementary Fig. 12, Supplementary Table 6). We then proceed to train RF classifiers (for each of those eight downsampled datasets) (Supplementary Fig. 13A), on 70% of the total amount of cells and kept aside 30% to test the performance of the classifier (Supplementary Fig. 13B). The red line in Fig. 3a and Supplementary Fig. 14 very clearly illustrates how the performance of the RF classifier, measured through the F1 score, gradually decreases from 0.95 to 0.45 while the downsampling rate goes from 50% to 0.5%. To see if we could mitigate this deterioration, we tested two ways of augmenting our alternate datasets. First, we used a naïve method, which we call upsampling, where we simply enlarged the cluster 2 population by duplicating the cells that were left after the downsampling procedure (Supplementary Fig. 13A). The orange line in Fig. 3a shows that this naive strategy actually mitigates the effect of the downsampling, albeit only to a minor extent (F1 score of 0.6 obtained for a downsampling rate of 0.5%). It is important to note that adding noise (e.g. standard Gaussian) to the upsampled cluster 2 cells usually deteriorated the classification performance (data not shown).

In order to understand whether in silico-generated cluster 2 cells could improve the RF performance, we next trained the cscGANs on the eight downsampled datasets (Supplementary Fig. 13C). We then proceeded to augment the cluster 2

population with the cells generated by the cscGAN (Supplementary Fig. 13A). Figure 3c shows that using as little as 2% (301 cells) of the real cluster 2 data for training the cscGAN suffices to generate cells that overlap with real test cells. When less cells are used the t-SNE overlap of cluster 2 training cells and generated cells slightly decreases (Fig. 3d, Supplementary Fig. 15). These results strongly suggest that the cluster-specific expression and gene dependencies are learned by the cscGAN, even when very few cells are available. In line with this assumption, the blue curves in Fig. 3a and Supplementary Fig. 14 show that augmenting the cluster 2 population with cluster 2 cells generated by the cscGAN almost completely mitigates the effect of the downsampling (F1 score of 0.93 obtained for a downsampling rate of 0.5%). We obtained similar results with RFs that have been optimized for the number of trees and features per tree (Supplementary Fig. 14D), showing that augmentation robustly increases classification performance across RF hyper-parameter space. Interestingly, the RF improves with increasing numbers of generated cells used for the classifiers' training (Supplementary Fig. 16).

Two conclusions can be obtained from these results. First the obvious, few cluster-specific cells do not represent the population well. Second, the usage of cscGAN-generated scRNA-seq data can mitigate this effect and increases the performance of downstream applications like classification when limited samples of a specific cluster are available.

**Improved trajectory analysis using augmented data**. The previous results highlight the ability of the cscGAN to specifically generate cells corresponding to different types or clusters. Such discrete states, however, are not sufficient to capture intermediate and transitional cellular states of an organism. Erythrocytes, for example, are derived in the red bone marrow from pluripotent stem cells that give rise to all types of blood cells. This differentiation process contains transitional cellular states that can be visualized (Supplementary Fig. 17A–C) using a pseudo-time analysis of bone marrow scRNA-seq data[28] (see Supplementary Table 1, Methods). The outcome of pseudo-time analyses, however, depends heavily on how well the variety of continuous states of erythrocytes is represented in the data. To highlight this property, we manually downsampled a subpopulation of erythrocytes in the bone marrow dataset. We can observe in Supplementary Fig. 17D–F that such downsampling directly affects the structure of the graph inferred by the pseudo-time analysis.

To show that the scGAN can reliably model populations that exist in continuous cellular states, we trained it on the downsampled bone marrow dataset. We then replaced the cells that were re-moved from the original data with handpicked scGAN-generated cells that belonged to the same subpopulation of erythrocytes. Adding the cells generated by scGAN allows to restore the original structure of the graph (Supplementary Fig. 17G–I). These results suggest that scGANs are able to model discrete and continuous cellular states and cell trajectories.

**cscGAN learns and translates gene-regulatory syntax**. The fidelity with which the (c)scGAN creates cells of very sparse populations is striking and it is tempting to speculate if the model actually learns and translates gene-regulatory information from abundant cell clusters to sparse ones.

We trained scGANs on decreasing amounts of cluster 2 cells (keeping 50%, 25%, 10%, 5%, 3%, 2%, 1%, 0.5%, 0.2%, and 0.1% of the initial population) and compared scGAN-generated cluster 2 cells to real test cluster 2 cells. In addition, we trained cscGANs on the same number of cluster 2 cells and all other clusters (see also previous section). We then compared scGAN (trained only

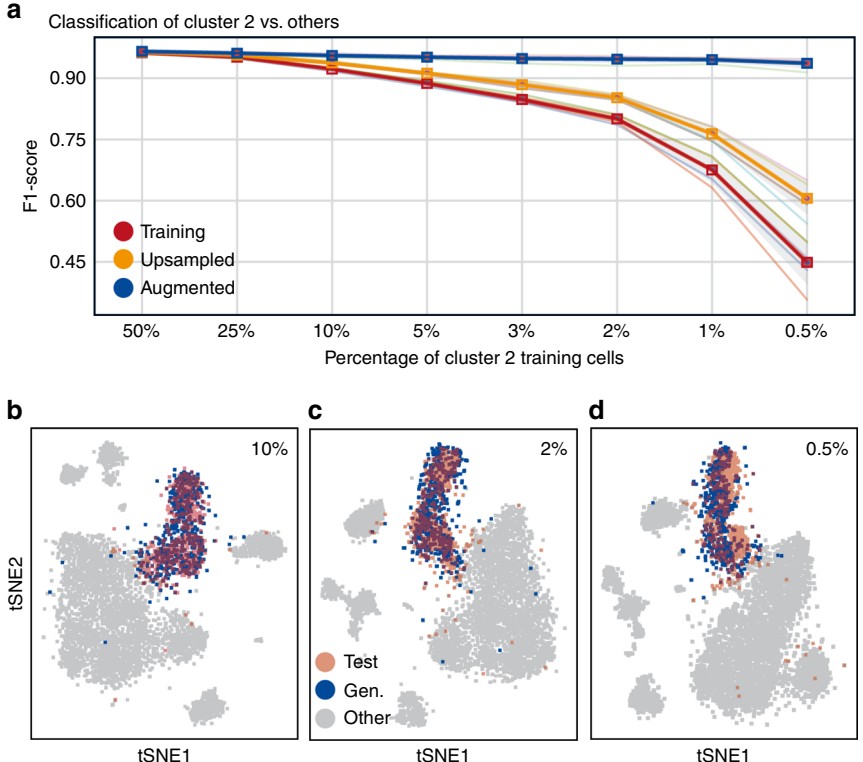

**Fig. 3 Effects of data downsampling and augmentation on classification and clustering. a** F1 score of an RF classifier trained to discriminate cluster 2 test from other test cells when trained on training (red), upsampled (orange), or augmented (blue) cells for eight levels of downsampling (50% to 0.5%). **b–d** t-SNE representation of cluster 2 real test (red) and cscGAN-generated (blue) cells for three levels of downsampling (10% panel **b**, 2% panel **c**, and 0.5% panel **d**). Other test cells are shown in gray.

on cluster 2) and cscGAN (trained on all clusters) generated cluster 2 cells to real test cluster 2 cells using RF classification and t-SNE visualization. The underlying hypothesis is that if the cscGAN can learn and translate general rules of gene regulation from abundant to sparse cell populations, it should provide more realistic cells for sparse clusters than a scGAN that was only trained on the latter.

We first assessed model fidelity by the ability of an RF classifier to distinguish scGAN and cscGAN-generated cluster 2 cells from real test cluster 2 cells. While the scGAN trained on a large number of cluster 2 cells generates more realistic cluster 2 cells than a cscGAN trained on all cell clusters, the cscGAN generates realistic cells of much wider variety than the scGAN when only few cluster 2 training cells are available (Supplementary Fig. 18). While the cscGAN seems to leverage gene-regulatory information from the more abundant clusters to compensate for the missing cluster 2 observations, the scGAN seems to re-create the few cluster 2 cells it has learned from, failing to generalize to unseen cluster 2 test cells (Supplementary Fig. 18C, D).

These results suggest that (c)scGAN can learn fundamental gene-regulatory rules that are valid across the observed cells (clusters and types). The (c)scGAN seems to learn those rules from the cells of large clusters and might apply them when generating cells of very small cell clusters.

## Discussion

This work shows how cscGAN can be used to generate realistic scRNA-seq representations of complex scRNA-seq data with multiple distinct cell types and millions of cells. cscGAN outperforms current methods in the realistic generation of scRNA-seq data and scales sublinearly in the number of cells. Most importantly, we provide compelling evidence that generating in

silico scRNA-seq data improves downstream applications, especially when sparse and underrepresented cell populations are augmented by the cscGAN-generated cells. We specifically show how the classification of cell types can be improved when the available data are augmented with in silico-generated cells, leading to classifiers that rival the predictive power of those trained on real data of similar size.

It may be surprising or even suspicious that our cscGAN is able to learn to generate cells coming from very small subpopulations (e.g. 16 cells) so well. We speculate that although cells from a specific type may have very specific functions, or exhibit highly singular patterns in the expression of several marker genes, they also share a lot of similarities with the cells from other types, especially with those that share common precursors. In other words, the cscGAN is not only learning the expression patterns of a specific subpopulation from the (potentially very few) cells of that population, but also from the (potentially very numerous) cells from other populations. This hypothesis actually aligns with the architecture of the cscGAN. In the generator, the only parameters that are cluster specific are those learned in the Conditional Batch Normalization layers (BLN). On the other hand, all the parameters of each of the Fully Connected (FC) layers are shared across all the different cell types.

While focusing on the task of cell type classification in this manuscript, many other applications will most probably gain from data augmentation, including—but not limited to—clustering itself, cell type detection, and data denoising. Indeed, a recent manuscript used Wasserstein GANs (WGAN) to denoise scRNA-seq data[24]. For this purpose, the (low-dimensional) representation obtained at the output of the single hidden layer of a critic network was used. These lower-dimensional representations keep cell type-determining factors while they discard noisy

information such as batch effects. In general, GAN models allow for the simulation of cell development or differentiation through simple arithmetic operations applied in the latent space representation of the GAN, operations for which our conditional cscGAN is especially suited.

Throughout this manuscript, we solely focused on using cell types as a side information to condition the generation on. It is worth mentioning that any other kind of side information (partitioning of the sample) could equally be used. For instance, a cscGAN could be conditioned and trained on a combination of case and control samples. While many other choices could lead to interesting applications, we leave this avenue of research for future work.

It is tempting to speculate how well the scRNA-seq data generation using cscGAN can be applied to other biomedical domains and data types. It is easy to envision, for example, how cscGAN variants could generate realistic (small) RNA-seq or proteomic data. Moreover, cscGAN variants might successfully generate whole genomes with predefined features such as disease state, ethnicity, and sex, building virtual patient cohorts for rare diseases, for example. In biomedical imaging, in silico image generation could improve object detection, disease classification, and prognosis, leading to increased robustness and better generalization of the experimental results, extending clinical application.

We hypothesize that data augmentation might be especially useful when dealing with human data, which is notoriously heterogeneous due to genetic and environmental variation. Data generation and augmentation might be most valuable when working with rare diseases or when samples with a specified ethnicity or sex, for example, are simply lacking.

Lastly we would like to emphasize that the generation of realistic in silico data has far reaching implications beyond enhancing downstream applications. In silico data generation can decrease human and animal experimentation with a concomitant reduction in experimental costs, addressing important ethical and financial questions.

## Methods

**Datasets and preprocessing**. *PBMC*: We trained and evaluated all models using a published human dataset of 68,579 PBMCs (healthy donor A)[12]. The dataset was chosen as it contains several clearly defined cell populations and is of reasonable size. In other words, it is a relatively large and complex scRNA-seq dataset with very good annotation, ideal for the learning and evaluation of generative models.

The cells were sequenced on Illumina NextSeq 500 High Output with ~20,000 reads per cell. The cell barcodes were filtered as in ref. [12] and the filtered gene matrix is publicly available on the 10x Genomics website.

In all our experiments, we removed genes that are expressed in less than three cells in the gene matrix, yielding 17,789 genes. We also discarded cells that have less than 10 genes expressed. This, however, did not change the total number of cells. Finally, the cells were normalized for the library size by first dividing UMI counts by the total UMI counts in each cell and then multiplied by 20,000. See Supplementary Table 1 for an outlook of this dataset.

*Brain Large*: In addition to the PBMC dataset we trained and evaluated our best performing scGAN model on the currently largest available scRNA-seq dataset of ~1.3 million mouse brain cells (10x Genomics). The dataset was chosen to prove that the model performance scales to millions of scRNA-seq cells, even when the organism, tissue, and the sample complexity varies. The sequenced cells are from the cortex, hippocampus, and the subventricular zone of two E18 mice.

The barcodes filtered matrix of gene by cell expression values is available on the 10x Genomics website. After removing genes that are expressed in less than three cells, we obtained a gene matrix of 22,788 genes. We also discarded cells that have less than 10 genes expressed, which did not affect the overall number of cells. The cells were normalized for the library size as described in the PBMC section.

*Brain Small*: We also examined the performance of the generative models proposed in this manuscript on a subset of the Brain Large dataset provided by 10x Genomics, which consists of 20,000 cells. The preprocessing of the Brain Small dataset was identical to that of the Brain Large dataset, yielding a matrix of 17,970 genes by 20,000 cells (Supplementary Table 1).

*Bone Marrow*: In order to understand the ability of the scGAN to learn the distribution from imputed cells we used a mouse bone marrow cell dataset (GSE72857)[29]. The cells were collected using a plate-based MARS-seq protocol in order to identify myeloid progenitor subpopulations. The sparsity and the heterogeneity of cells in this dataset makes it suitable for imputation. Data preprocessing was performed as described above, yielding a matrix of 12,443 genes by 2,730 cells (Supplementary Table 1). Processed bone marrow cells were either used directly for scGAN modeling or after imputation using MAGIC (see section Expression imputation with MAGIC).

*Zeisel*: Finally, we trained scGANs on somatosensory cortex (S1) and hippocampal CA1 cells (GSE60361)[20], which consists of 3,005 high-quality single cells (including neurons, glia, and endothelial cells). After preprocessing we obtained a matrix of 18,738 genes by 3,005 cells (Supplementary Table 1).

*Clustering*: Throughout this manuscript we use the Cell Ranger workflow for the scRNA-seq secondary analysis[12]. First, the cells were normalized by UMI counts. Then, we took the natural logarithm of the UMI counts. Afterwards, each gene was normalized such that the mean expression value for each gene is 0, and the standard deviation is 1. The top 1000 highly variable genes were selected based on their ranked normalized dispersion. PCA was applied on the selected 1000 genes. In order to identify cell clusters, we used Louvain clustering[30] on the first 50 principal components of the PCA. This replaced the k-means clustering used in Cell Ranger R analysis workflow, as the Scanpy[31] tutorial on clustering the PBMC dataset advises. The number of clusters were controlled by the resolution parameter of scanpy.api.tl.louvain. The higher resolution made it possible to find more and smaller clusters.

For the PBMC and the Brain Large dataset we used a resolution of 0.15 which produced 10 and 13 clusters, respectively. The Brain Small dataset was clustered using a resolution of 0.1 which gives 8 clusters.

To understand if the selection of different clustering algorithms might affect the fidelity with which the cscGAN models scRNA-seq data, we compared the results obtained with Louvain clustering compare to that of K-means clustering on the PBMC dataset. We used the scikit-learn package[32] to apply the clustering on the first 50 principal components extracted as mentioned above. The K-means scikit-learn function default parameters were used for the clustering except for the number of centroids to generate in the data, which was set to 10 (the number of clusters previously obtained with the Louvain algorithm). This produces 10 clusters (Supplementary Fig. 11). The results obtained are very similar to those with the Louvain clustering in terms of the ability of an RF classifier to discriminate between real cells and cscGAN-generated cells (Supplementary Table 5).

*Definition of marker genes*: In several experiments we investigated the expression levels and correlation of genes. For this purpose, a group of 10 marker genes was defined by taking the five most highly upregulated genes for the largest two clusters in the dataset (clusters 1 and 2 for the PBMC dataset). Significant upregulation was estimated using the logarithm of the Louvain-clustered cells with the scanpy.api.tl.rank_genes_groups function with its default parameters (Scanpy 1.2.2)[31].

**Model description**. *scGAN*: In this section, we outline the model used for the scGAN by defining the loss function it optimizes, the optimization process, and key elements of the model architecture. GANs typically involve two Artificial Neural Networks: a generator, which, given some input random noise, trains to output realistic samples, and a critic that trains to spot the differences between real cells and the ones that the generator produces (Supplementary Fig. 1). An adversarial training procedure allows for those entities to compete against each other in a mutually beneficial way. Formally, GANs minimize a divergence between the distributions of the real samples and of the generated ones. Different divergences are used giving rise to different GAN variants. While original GANs[5] minimize the so-called Jensen–Shannon divergence, they suffer from known pitfalls making their optimization notoriously difficult to achieve[33]. For instance, they are known to be prone to mode collapse, where the generated samples are realistic albeit only representing a fraction of the variety of the samples it was trained on (i.e. only a few but not all modes of the distribution of the real samples is learned). On the other hand, WGANs[15,34] use a Wasserstein distance, with compelling theoretical and empirical arguments. In our hands, WGANs showed no evidence of mode collapse and showed stable and robust training with respect to hyper-parameter optimization. On a side note, early attempts to train an original GAN on scRNA-seq data never yielded convergence, while an out-of-the-box implantation of a WGAN did. This does not imply that it is impossible to successfully train an original GAN on such data.

Let us denote by $P_r$ and $P_s$ the distributions of the real and of the generated cells respectively. The Wasserstein distance between them, also known as the Earth Mover distance, is defined as follows:

$$W(P_r, P_s) = \inf_{\gamma \in \Pi(P_r, P_s)} \mathbb{E}_{(\mathbf{x},\mathbf{y}) \sim \gamma} ||\mathbf{x} - \mathbf{y}||, \qquad (1)$$

where $x$ and $y$ are random variables and $\prod(P_r, P_s)$ is the set of all joint distributions $\gamma(\mathbf{x}, \mathbf{y})$ whose marginals are $P_r$ and $P_s$, respectively. Those distributions represent all the ways (called transport plans) you can move masses from $x$ to $y$ in order to transform $P_r$ into $P_s$. The Wasserstein distance is then the cost of the optimal transport plan.

However, in this formulation, finding a generator that will generate cells coming from a distribution $P_s$ such that it minimizes the Wasserstein distance with the distribution of the real cells is intractable.

Fortunately, we can use a more amenable, equivalent formulation for the Wasserstein distance, given by the Kantorovich–Rubinstein duality:

$$W(P_r, P_s) = \sup_{||f||_L \leq 1} \mathbb{E}_{\mathbf{x} \sim P_r} f(\mathbf{x}) - \mathbb{E}_{x \sim P_s} f(\mathbf{x}), \tag{2}$$

where $||f||_L \leq 1$ is the set of 1-Lipschitz functions with values in $\mathbb{R}$. The solution to this problem is approximated by training a Neural Network that we previously referred to as the critic network, and whose function will be denoted by $f_c$.

The input of the generator are realizations of a multivariate noise whose distribution is denoted by $P_n$. As it is common in the literature, we use a centered Gaussian distribution with unit diagonal covariance (i.e. a multivariate white noise). The dimension of the used Gaussian distribution defines the size of the latent space of the GAN. The dimension of that latent space should reflect the intrinsic dimension of the scRNA-seq expression data we are learning from, and is expected to be significantly smaller than their apparent dimension (i.e. the total number of genes).

If we denote by $f_g$ the function learned by our generator network, the optimization problem solved by the scGAN is identical to that of a Wasserstein GAN:

$$\min_{f_g} \max_{||f_c||_L \leq 1} \mathbb{E}_{\mathbf{x} \sim P_r} f_c(\mathbf{x}) - \mathbb{E}_{\mathbf{x} \sim f_g(P_n)} f_c(\mathbf{x}). \tag{3}$$

The enforcement of the Lipschitz constraint is implemented using the gradient penalty term proposed by Gulrajani et al.[34].

Hence, training an scGAN model involves solving a so-called minmax problem. As no analytical solution to this problem can be found, we recourse to numerical optimization schemes. We essentially follow the same recipe as most of the GAN literature[5,33], with an alternated scheme between maximizing the critic loss (for five iterations) and minimizing the generator loss (for one iteration). For both the minimization and the maximization, we use a recent algorithm called AMSGrad[35], which addresses some shortcomings of the widely used Adam algorithm[36], leading to a more stable training and convergence to more suitable saddle points. The AMSGrad exponential decay parameter beta1 was set to 0.5 and beta2 to 0.9.

Regarding the architecture of our critic and generator networks, which is summarized in Supplementary Fig. 1, most of the existing literature on images prescribes the use of convolutional neural networks (CNN). In natural images, spatially close pixels exhibit stronger and more intricate inter-dependencies. Also, the spatial translation of an object in an image usually does not change its meaning. CNNs have been designed to leverage those two properties. However, neither of these properties hold for scRNA-seq data, for which the ordering of the genes is mostly arbitrary and fixed for all cells. In other words, there is no reason to believe that CNNs are adequate, which is why scGAN uses FC layers. We obtained the best results using an MLP with FC layers of 256, 512, and 1024 neurons for the generator and an MLP with FC layers of 1024, 512, 256 for the critic (Supplementary Fig. 1B, C). At the outermost layer of the critic network, following the recommendation from Arjovsky and Bottou[33], we do not use any activation function. For every other layer of both the critic and the generator networks, we use a Rectified Linear Unit (ReLU) as an activation function.

Naturally, the optimal parameters in each layer of the artificial neural network highly depends on the parameters in the previous and subsequent layers. Those parameters, however, change during the training for each layer, shifting the distribution of subsequent layer's inputs slowing down the training process. In order to reduce this effect and to speed up the training process, it is common to use Normalization layers such as Batch Normalization[37] for each training mini-batch. We found that the best results were obtained when using Batch Normalization at each layer of the generator. Finally, as mentioned in the Datasets and preprocessing section, each real sample used for training has been normalized for library size. We now introduce a custom LSN layer that enforces the scGAN to explicitly generate cells with a fixed library size (Supplementary Fig. 1B).

*LSN layer*: A prominent property of scRNA-seq is the variable range of the genes expression levels across all cells. Most importantly, scRNA-seq data are highly heterogeneous even for cells within the same cell subpopulation. In the field of Machine Learning, training on such data is made easier with the usage of input normalization. Normalizing input yields similarly ranged feature values that stabilize the gradients. scRNA-seq normalization methods that are used include LSN, where the total number of reads per cell is exactly 20,000 (see also Datasets and preprocessing).

We found that training the scGAN on library-size normalized scRNA-seq data helps the training and enhances the quality of the generated cells in terms of our evaluation criteria (model selection method). Providing library-size normalized cells for training of the scGAN implies that the generated cells should have the same property. Ideally, the model will learn this property inherently. In practice, to speed up the training procedure and make training smoother, we added the aforementioned LSN layer at the output of the generator (Supplementary Fig. 1B). Our LSN Layer rescales its inputs ($\bar{\mathbf{x}}$) to have a fixed, total read count ($\varphi$) per cell:

$$\mathbf{y}_{relu} = \text{ReLU}(\bar{\mathbf{x}}W + \mathbf{b}), \tag{4}$$

$$\mathbf{y}_{output} = \frac{\varphi}{\sum_i (\mathbf{y}_{relu})_i} \mathbf{y}_{relu}, \tag{5}$$

where $W$ and $\mathbf{b}$ are its weights and biases, and $(\mathbf{y}_{relu})_i$ denotes the $i$th component of the $\mathbf{y}_{relu}$ vector.

*cscGAN*: Our cscGAN leverages conditional information about each cell type, or subpopulation, to enable the further generation of type-specific cells. The integration of such side information in a generative process is known as conditioning. Over the last few years, several extensions to GANs have been proposed to allow for such conditioning[26,38,39]. It is worth mentioning that each of those extensions are available regardless of the type of GAN at hand.

We explore two conditioning techniques, auxiliary classifiers (ACGAN)[39] and projection-based conditioning (PCGAN)[26]. The former adds a classification loss term in the objective. The latter implements an inner product of class labels at the critic's output. While we also report results obtained with the ACGAN (see Supplementary Table 4), the best results were obtained while conditioning through projection.

In practice, the PCGAN deviates from the scGAN previously described by (i) multiple critic output layers, one per cell type and (ii) the use of Conditional BNL[27], whereby the learned singular scaling and shifting factors of the BNL are replaced with one per cell type.

As described in Section 2 and 3 of ref.[26], the success of the projection strategy relies on the hypothesis that the conditional distributions (with respect to the label) of the data at hand are simpler, which helps stabilizing the training of the GAN. When it comes to scRNA-seq data, it is likely that this hypothesis holds as the distribution of the gene expression levels should be simpler within specific cell types or subpopulations.

**Model selection and evaluation**. Evaluating the performance of generative models is no trivial task[13,14]. We designed several metrics to assess the quality of our generated cells at different levels of granularity. We will now describe in detail how those metrics were obtained. They can be grouped into two categories: the metrics we used for model selection (in order to tune the hyper-parameters of our GANs) and the metrics we introduced in the Results section.

**Metrics used for model selection**. As described in the previous section, defining our (c)scGAN model entails carefully tuning several hyper-parameters. We hereby recall the most influential ones: (i) the number and size of layers in the Neural Networks, (ii) the use of an LSN layer, and (iii) the use of a Batch Normalization in our generator network.

For each of our models, before starting the training, we randomly pick 3000 cells from our training data and use them as a reference to measure how it performs. We therefore refer to those 3,000 cells as "real test cells".

To optimize those hyper-parameters, we evaluated various models and evaluated their performance through a few measures, computed during the training procedure: (a) the distance between the mean expression levels of generated cells and real test cells, (b) the mean sparsity of the generated cells, and (c) the intersection between the most highly variable genes between the generated cells and the real test cells.

First, we compute the mean expression value for each gene in the real test cells. During the training procedure, we also compute the mean expression value for each gene in a set of 3,000 generated cells. The discrepancy is then obtained after computing the Euclidean distance between the mean expression values of the real and the generated cells.

scRNA-seq data typically contains a lot of genes with 0 read counts per cell, which we also use to estimate the similarity of generated and real cells. Naturally, similar sparsity values for real and test cells indicate good model performance whereas big differences indicate bad performance.

Finally, using the Scanpy[31] package, we estimate the 1000 most highly variable genes from the real data. During the training, we also estimate what are the 1,000 most highly variable genes from a sample of 3,000 generated cells. We use the size of the intersection between those two sets of 1,000 highly variable genes as a measurement of the quality of the generation.

**Gene expression and correlation**. To highlight the performance of our models, we used violin plots of the expression of several marker genes along with heatmaps displaying the correlation between those same marker genes as expressed among all clusters, or among specific clusters.

To produce those plots, we used the expression levels of cells (either test real, or generated by scGAN, cscGAN) in a logarithmic scale. For the heatmaps we compute the Pearson product–moment correlation coefficients.

**t-SNE plots**. To visualize generated and real cells within same t-SNE plot they are embedded simultaneously. In brief, we are trying to assess how realistic the generated cells are. Thus our reference point is the real data itself. The delineation of what constitutes noise and what constitutes biologically relevant signal should be driven by the real data only. Hence we project the generated cells on the first 50 principal components that were computed from the real cells in the Cell Ranger pipeline[12] (see also Datasets and preprocessing). From this low-dimensional representation, we compute the t-SNE embedding.

To show that the results we obtained were not an artifact of using a Principal Components Analysis, we also reported (Supplementary Fig. 8) the results (t-SNE plots and classification results) obtained while using the first 50 components of

ZIFA[40] (Zero-Inflated Factor Analysis), computed on both the real and generated cells, as an alternate dimensionality reduction method.

**Classification of real versus generated cells.** Building on the 50-dimensional representation of the cells (t-SNE plots section), we trained classifiers to distinguish between real test cells and generated cells. Using this lower-dimensional representation is motivated by the fact that it captures most of the biologically relevant information while discarding most of the noise, which is known to be high in scRNA-seq data. Moreover, it is statistically more sound to use a dimensionality reduction technique prior to classifying data when the number of observations is in the same order of magnitude as the number of variables, as is the case with the datasets we worked with. As mentioned in the Results section, we trained RF classifiers with 1000 trees and a Gini impurity quality metric of the decision split using the scikit-learn package[32]. The maximum depth of the classifier is set so that the nodes are expanded until all leaves are pure or until all leaves contain less than two samples. The maximum number of features used is the square root of the number of genes.

In order to produce Fig. 1e, which highlights the ability to separate real from generated cells, irrespective of which cluster they are coming from, we used the whole real test set along with generated cells. On the other hand, Fig. 2g, h is cluster specific (cluster 2 and cluster 5 respectively). We trained the RFs using only the cells from those specific clusters. To prevent bias due to class imbalance, each model was trained using an equal number of real test cells and generated cells.

We used a five-fold cross-validation procedure to estimate how each classifier generalizes. To assess this generalization performance, we plotted the Receiver Operating Characteristic (ROC) curves obtained for each fold, along with the average of all the ROC curves. We also display the AUC in each of those cases. Supplementary Tables 4 and 5 report more extensive results.

**MMD distance.** Computing robust distances over empirical distributions is a difficult issue in high dimension. However, a recent framework called kernel two-sample test[41] was proposed as a statistical test to assess whether two samples are coming from the same distribution. It relies on the computation of a distance called MMD. In a nutshell, it compares the first-order moments (means) of the two samples, in a reproducing kernel Hilbert space. As a consequence, the choice of the kernel is of paramount importance.

Following the recommendations from Shaham et al.[42], which uses a deep neural network to minimize the MMD distance between different scRNA-seq data replicates for batch effect removal, we used a kernel that is the sum of three Gaussian kernels:

$$k(\mathbf{x}, \mathbf{y}) = \sum_i \exp\left(-\frac{\|\mathbf{x} - \mathbf{y}\|^2}{\sigma_i^2}\right),$$

where $\sigma_i$'s are chosen to be $\frac{m}{2}$, $m$, $2m$ and $m$ is the median of the average distance between a point to its nearest 25 neighbors.

For the sake of consistency with the other measures we proposed (t-SNE plots, RF classification), we proceeded to compute the MMD distances between samples using their 50 PCs representation found with the Cell Ranger pipeline (as described in the "Dataset and preprocessing" part).

We used the MMD implementation from SHOGUN[43], an efficient kernel-based machine learning package.

**Downsampling.** To assess the impact of cluster size on the ability of the cscGAN to model the cluster we artificially reduced the number of cells of the relatively large PBMC cluster 2. We call this approach "downsampling" throughout the manuscript.

Eight different percentages {50%, 25%, 10%, 5%, 3%, 2%, 1%, 0.5%} of cluster 2 cells were sampled using a random seed and a uniform sampling distribution over all the cluster 2 cells (Supplementary Table 5). We sampled nested subsets (for each seed, the smaller percentage samples are a complete subset of the larger ones). In order to accurately estimate the generalization error across the different experiments and to avoid potential downsampling artifacts, we conducted all our experiments using five different random seeds. For the classification of cell subpopulations (see next paragraph) we report the average F1 score as well as the five individual F1 score values for the different seeds.

**Classification of cell subpopulations.** To investigate the use of the proposed cscGAN model for data augmentation, we examined the performance of cell subpopulation classification before and after augmenting the real cells with generated cells. For this purpose, and as described in the previous paragraph and the Results section, we produced alternate datasets with sub-sampled cluster 2 populations (Supplementary Fig. 12).

For simplicity, we focus in this section on the experiment where cluster 2 cells were downsampled to 10% using five different random seeds (Supplementary Fig. 13). We advise to use Supplementary Fig. 13 as an accompanying visual guide to this text description.

Using the previously introduced 50-dimensional PC representation of the cells, three RF models were trained to distinguish cluster 2 cells from all other cell populations (RF downsampled, RF upsampled, and RF augmented) (Supplementary Fig. 13A). In the training data for all the three classifiers, 70% of the cells from all the clusters except cluster 2 (i.e. 37,500 cells) were used (light blue boxes in Supplementary Fig. 13A).

*RF downsampled*: For the first RF classifier, we used 10% of cluster 2 cells (1502 cells) and 70% other cells (37,500 cells) to train the RF model. We refer to this dataset as the "RF downsampled" set in Supplementary Fig. 13A. This dataset was also used to train the cscGAN model, which is used later to generate in silico cluster 2 cells (Supplementary Fig. 13C). It is important to note that RF classifiers for "RF downsampled" datasets always use weights that account for the cluster-size imbalance. The reason for this is that RFs are sensitive to unbalanced classes, leading to classifiers that always predict the much larger class, thereby optimizing the classification error[44].

*RF upsampled*: For the second RF classifier, we uniformly sampled with replacement 5,000 cells from the 1,502 cluster 2 cells (10%). We added those 5,000 (copied) cells to the original 1,502 cluster 2 cells. This dataset is referred to as "RF upsampled" in Supplementary Fig. 13A. The rationale for this upsampling is that RF multinomial classifiers are sensitive to the class frequencies in the training data. The upsampling was conducted only for cluster 2 cells as a baseline to which the augmentation is compared. As the augmented and upsampled datasets remain unbalanced, we adjusted the class weights during the training to be inversely proportional to the class frequencies, as outlined in the previous paragraph (RF downsampled).

As a side note, we also conducted experiments where we added standard Gaussian noise to the upsampled cells, which always reduced the performance of the RF classifier and are therefore not shown.

*RF augmented*: Finally, the third classifier training data "RF augmented" consists of 10% cluster 2 cells as well as 5,000 cluster 2 cells generated using the 10% cscGAN model as shown in Supplementary Fig. 9C. The 10% cscGAN model was trained on 10% cluster 2 cells as well as all other cells (53,571 cells, Supplementary Fig. 13C).

The RF classifiers were trained using the same parameters as described in the Classification of real versus generated cells methods section, using 1,000 trees and Gini impurity. The only difference is that here the class weights during the training are adjusted inversely proportional to the class frequencies, as already mentioned above. The scikit-learn package[32] was used to conduct all experiments to classify cell subpopulations.

*Test cells*: The test cells used to evaluate the classifiers consisted of 30% of the data from all the clusters. Since we are testing the cscGAN's ability to augment different percentages of real cluster 2 cells, we made sure that the 30% of cluster 2 cells used in the test set were selected from the cells which were not seen by any trained cscGAN model (Supplementary Fig. 13B).

To prove that the downsampling limits the ability to classify and that augmenting the dataset mitigates this effect, all three RF classifiers were trained to classify cluster 2 cells versus all other subpopulations. The F1 score of each classifier is calculated and presented in different colors (Fig. 3a).

Furthermore, in order to understand how augmentation helps to separate close clusters, we trained the same three RF classifiers after removing all clusters except cluster 2 and 1 from the corresponding training data. We repeated this procedure for cluster 2 and 5, and cluster 2 and 3. We chose those clusters in particular because their highly differentially expressed genes are also highly expressed in 2 meaning that separating them from cluster 2 is more difficult (Fig. 1a–c). In a similar way, F1 scores for classification of cluster 2 versus 1 (Supplementary Fig. 14A), cluster 2 versus 3 (Supplementary Fig. 14B), and cluster 2 versus 5 (Supplementary Fig. 14C) are calculated and reported.

As mentioned above, we repeated this procedure for different downsampling levels of cluster 2 cells and for five different sampling seeds for each level (Supplementary Table 5).

When training the RF classifier with the augmented dataset, the number of cells used in the augmentation was set to 5,000 cells. This, however, does not necessarily mean that 5,000 cells is the optimal number of cells to be added. The increase in the F1 score due to augmenting the data with generated cells depends on two factors: (i) the number of real cells in the original subpopulation and (ii) the number of cells used for augmentation. To highlight the impact of the number of generated cells used for data augmentation, we trained the previously mentioned RF classifiers using different numbers of generated cells (from 100 to 12,000) while keeping the number of other cells constant (Supplementary Fig. 16).

**Splatter comparison.** In addition to what has been previously introduced in the Results section, we also compared the performance of the scGAN to Splatter[22], using the metrics described in their manuscript. Briefly, Splatter simulation is based on a gamma-Poisson hierarchical model, where the mean expression of each gene is simulated from a gamma distribution and cell counts from a Poisson distribution. We noticed that Splatter uses the Shapiro–Wilk test to evaluate the library-size distribution, which limits the number of input cells to 5,000. Therefore, we slightly modified the code that allows Splatter to take more than 5,000 cells as input.

While scGAN learns from and generates library-size normalized cells, Splatter is not suited for that task. For the sake of fairness, we used the Splatter package on the non-normalized PBMC training dataset. We then generated (non-normalized)

cells, which we normalized, so that they could be compared to the cells generated by scGAN. Following ref. [22], we used the following evaluation metrics: distribution of the mean expression, of the variance, of the library sizes and ratio of zero read counts in the gene matrix. The results were computed using the Splatter package and are reported in Supplementary Fig. 5.

We observe that the results obtained by Splatter are marginally better than or identical to these of scGAN (Supplementary Fig. 5). The results from those measures suggest that both Splatter and scGAN constitute almost perfect simulations. However, Splatter simulates virtual genes. While those genes share some characteristics with the real genes Splatter infers its parameters from, there is no one-to-one correspondence between any virtual gene simulated in Splatter-generated cells and the real genes. We therefore did not compare Splatter-simulated cells with real cells, as we did to evaluate the quality of (c)scGAN-generated cells. This also prohibits the use of Splatter for data-augmentation purposes.

This being said, we also would like to pinpoint that while the (c)scGAN is able to capture the gene–gene dependencies expressed in the real data (Fig. 1d, Supplementary Fig. 10), this does not hold for Splatter, for which the virtual genes are mostly independent from each other. To prove this point, we extract the 100 most highly variable genes from the real cells, the cells generated by Splatter, and the cells generated by the scGAN. We then proceed to compute the Pearson correlation coefficients between each pair within those 100 genes (Supplementary Fig. 4). It reveals that while those most highly variable genes in the real cells or those generated by the scGAN exhibit some strong correlations, highly variable genes are mostly independent from each other in the cells generated by Splatter. These results are surprising given that the graphical model used in Splatter is expressive enough to accommodate for complex dependencies between genes. It is likely that it is the inference algorithm that is failing at capturing the gene–gene dependencies in the PBMC dataset, while a manual selection of the parameters of Splatter can allow to simulate cells with some gene–gene dependencies.

**SUGAR comparison**. Another generative model of high-dimensional data that could be used to generate scRNA-seq data is SUGAR (Synthesis Using Geometrically Aligned Random-walks).

Both scGAN and SUGAR share the assumption that the training data lie on a low-dimensional manifold which is the case of single-cell data[23]. The scGAN uses a random variable $\mathbf{Z}$ with a fixed distribution $P(z)$ and passes it through a neural network based parametric function (the generator) $(\theta)\mathbf{z} \rightarrow \mathbf{x}$. The output of this parametric function $P_\theta$ is then learned using an Earth mover distance to be closer to the real distribution $P_r$. SUGAR, on the other hand, uses a Gaussian kernel to construct the diffused geometry around each data point and then, using a sparsity-based measure, new points are sampled to even out the sparsity along the manifold.

In order to compare the quality of scGAN-generated cells with SUGAR-generated cells, we run SUGAR on a group of training cells from the PBMC dataset using the publicly available MATLAB implementation (https://github.com/KrishnaswamyLab/SUGAR). The training cells were the same cells used to train an scGAN model and were preprocessed as described in Datasets section PBMC.

SUGAR could generate points to explicitly balance the density over the learnt manifold by assuming that there are sparse regions and then generating points to equalize the estimated sparse areas on the learnt manifold. However such an equalization produces cells that, by design, do not follow the original distribution of the real cells. Therefore, we turned off the density equalization option when we generated cells using SUGAR to ensure that the generated cells compare favorably to the real ones in terms of distribution. For the same reason, we also turned off the imputation step. Finally, using the adaptive noise covariance estimation option of SUGAR resulted in scalability issues (Supplementary Fig. 6F–I, training and generating cells on a reduced 3,000 cells × 2,000 genes dataset required 1.3 Terabytes of RAM and computed for over 36 h), precluding the use of this option on the PBMC dataset. Following SUGAR co-author suggestions, we fixed the noise covariance matrix to be the identity matrix in order to allow SUGAR to generate cells in the original genes space using the available MATLAB version. The generated cells using SUGAR contained some negative values which we replaced with zeros to comply with our analysis workflow (logarithmic transformation using Cell Ranger). For the visualization of the SUGAR-generated cells, we used t-SNE to obtain a two-dimensional visualization of the generated and the real cells (Supplementary Fig. 6A–C). We also computed the MMD statistic obtained from the comparison of real (test) data with the generated data using both SUGAR (59.45) and scGAN (0.872) as described in the MMD methods (Supplementary Table 3). It is worth noting that while the Gaussian noise, added to the real cells, is the crux of how SUGAR generates novel cells. It, however, also may be the reason why the samples produced by SUGAR do not follow the original distribution of the data as closely as those produced by scGAN.

To investigate whether the gene–gene dependencies were kept in the SUGAR-generated data we computed the Pearson correlation coefficients of the cluster-specific marker genes (Supplementary Fig. 6D).

Lastly, we trained an RF classifier to distinguish between the real and the SUGAR-generated cells. We conjectured that RF classifier should have close to chance-level performance in the task of distinguishing the generated data from the real data. The RF classifier reaches 0.98 AUC when discriminating between the real and generated cells (blue curve in Supplementary Fig. 6E).

**Expression imputation with MAGIC**. An important aspect of using an scGAN for generating realistic cells is its fidelity in learning the distribution of the input data regardless of the preprocessing which is applied. Imputation of scRNA-seq data is used to denoise the data, to reduce the amount of drop-outs, and consequently to more accurately recover the gene–gene interactions. For this reason, we investigated the ability of an scGAN to generate realistic imputed cells when real imputed cells are used in the training.

We used MAGIC[25] to impute the scGAN training data, a method developed to impute missing values and to restore the structure of the scRNA-seq data.

The Mouse Bone Marrow dataset was used in this analysis after applying the basic filtering and the LSN we applied in all our experiments (refer to Datasets Supplementary Table 1, preprocessing section). The preprocessed cells are then imputed using the open source MAGIC implementation. In accordance with the MAGIC tutorial all genes were used with four diffusion steps. Afterwards, we trained an scGAN models for 100k steps on both imputed and non-imputed data. Both models were used to generate cells which we used to plot the gene–gene relationships of three genes in the form of scatter plots. To evaluate imputation fidelity we used the three genes that were used in the MAGIC online tutorial (Ifitm1, a stem cell marker, Klf1, an erythroid marker, and Mpo, a myeloid marker).

**Regulon detection using SCENIC**. We used SCENIC[19] to evaluate whether scGANs model active regulons in the Zeisel RNA-seq dataset. This dataset was used by the authors of SCENIC to show cross-species Dlx1 regulon activity. We selected the top 50 target genes with highest weight for each TF and subsequently found significantly over-represented TF-binding motifs in the set of genes. Modules with enriched TF-binding motifs were kept and defined as active regulons. We then trained an scGAN model on the Zeisel dataset and used it to generate 10,000 library-size normalized cells. The Dlx1 regulon was then found in the real dataset (realDlx1) as well as in the generated one (genDlx1). In addition, we used AUCell to calculate the regulon binarized activity of the realDlx1 regulon in the cells of the generated dataset and the genDlx1 regulon in real cells. Reciprocal activity of realDlx1 and genDlx1 regulons are visualized using t-SNE on real and generated data (Supplementary Fig. 4).

**Pseudo-time analysis with PAGA**. In this analysis we investigate the ability and the fidelity of the scGAN model to generate scRNA-seq data corresponding to continuous cell states. For this purpose, we used PAGA pseudo-time topology-preserving embedding with partition-based graph abstraction. While clustering enables understanding the biological signals within cell populations, trajectory analysis using pseudo-time and graph embeddings allows for the interpretation of continuous phenotypes and processes such as development and disease progression[28]. We chose an scRNA-seq hematopoiesis dataset (bone marrow dataset) that contains many intermediate and transition states to investigate the performance of such trajectory analysis[29].

In order to examine the ability of scGAN models to learn the manifold the data lie on, we performed a pseudo-time analysis as described in the official git repository of the hematopoiesis scRNA-seq data [https://github.com/theislab/paga]. The cells were first preprocessed using the Zheng preprocessing pipeline. Afterwards the force-directed single-cell graph was built using 20 PCA components[45,46]. The graph is then de-noised and rebuilt using PAGA-initialization as described in the official tutorial of PAGA. Supplementary Figure 17A–C shows the graph of the bone marrow scRNA-seq data. Scanpy pseudo-time analysis was used to infer the progression of cells through geodesic distance along the graph[47].

In the next step, we downsampled a specific transient cell state represented by the cells grouped in node 4 of the PAGA graph (Supplementary Fig. 17C–F). The original fourth Louvain group population of 150 cells was downsampled to 13 cells. The downsampled scRNA-seq data (the training data) were then used to train an scGAN model without providing any prior information about the cells' states or clusters. After training, the model was used to generate cells that we compared to the original dataset. After investigating the force-directed single-cell graph of the generated cells combined with original cells, we noticed that the generated cells were covering all cellular states of the original scRNA-seq data (Supplementary Fig. 17C–F).

These results motivated us to investigate the fidelity of the scGAN to learn the downsampled cellular state of transient state 4. Therefore, we searched within the generated cells for cells that are close to the sparse area created by the downsampling process. A group of 137 generated cells were found and added to the downsampled scRNA-seq data. We refer to this combined group of generated and downsampled cells as augmented cells. Our assumption is that the generated cells recover the lost biological signal represented by the downsampled transient state. To prove this assumption, we plotted the force-directed single-cell graph of the augmented data and compared it with the one built from the downsampled data. The cells' graph embeddings were recomputed using PAGA-initialization so that the cells are structured in a meaningful topology-preserving layout that reflects the real cell–cell interconnections and the paths of single cells. Data augmentation of the downsampled cells re-established the developmental trajectories that were observed in the real data and lost in the downsampled data, as shown in

Supplementary Fig. 17. Of note, the scGAN was trained with reduced neuron numbers to accommodate for the small size of the dataset.

**Software, packages, and hardware used**. For the sake of reproducibility, here is a list of the version of all the packages we used: Tensorflow v1.8, Scanpy v1.2.2, Anndata v0.6.5, Pandas v0.22.0, Numpy v1.14.3, Scipy v1.1.0, Scikit-learn v0.19.1, R v3.5.0 (2018-04-23), loomR v0.2.0, SHOGUN v6.1.3, SingleCellexperiment v1.2.0, Splatter v1.4.0, SUGAR v0.0, MAGIC v1.3.0, SCENIC v0.1.7, GENIE3 v1.0.0, Rcistarget v0.99.0, AUCell v0.99.5, RcisTarget.mm9.motifDatabases.20k v0.1.1, ZIFA v0.1. Regarding hardware, all (c)scGAN models were trained on a single-GPU of an NVIDIA DGX-1 server (Tesla V100 GPUs).

**Reporting summary**. Further information on research design is available in the Nature Research Reporting Summary linked to this article.

## Data availability

The datasets used and analyzed during the current study are available on the 10x Genomics dataset repository at https://support.10xgenomics.com/single-cell-gene-expression/datasets/1.1.0/fresh_68k_pbmc_donor_a for PBMC, at https://support.10xgenomics.com/single-cell-gene-expression/datasets/1.3.0/1M_neurons for Brain small and Brain large, and in the Gene Expression Omnibus repository, at https://www.ncbi.nlm.nih.gov/geo/query/acc.cgi?acc=GSE72857 for Bone Marrow, and https://www.ncbi.nlm.nih.gov/geo/query/acc.cgi?acc=GSE60361 for Zeisel.

## Code availability

Our (c)scGAN Tensorflow[48] implementation can be found on https://github.com/imsb-uke/scGAN, including documentation for the training of the (c)scGAN models. As mentioned before, we used Scanpy[31] to conduct most of the data analysis. We also compared our results to those of Splatter[22], and adapted the code they provided on Github (https://github.com/Oshlack/splatter).

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

## Acknowledgements

We would like to thank Ulf Panzer and the Institute of Medical Systems Biology for helpful discussions and suggestions. In particular, we would like to thank Sven Heins and the ZMNH IT for setting up the Deep Learning IT infrastructure and the daily support. This work was supported by the grants SFB 1286/Z2 to P.M. and M.M.; SFB 1192 to C.K., C.F.K., and S.B.; DFG BO 4224/4-1 to D.S.M.; and BMBF IDSN to V.B.

## Author contributions

S.B. initiated the project. S.B., P.M., M.M. and D.S.M. designed the study, deep learning models, and analysis. P.M. M.M. and D.S.M. built the deep learning models. P.M., M.M., V.B., and C.K. analyzed the data. S.B., P.M., M.M., D.S.M., V.B., and C.F.K. contributed to the manuscript.

## Competing interests

The authors declare no competing interests.
