## [Peer Review File · Nature Communications]

Reviewers' comments:

Reviewer #1 (Remarks to the Author):

This paper proposes augmenting or generating single-cell RNA sequencing data using a conditional GAN. GANs are neural networks that use two adversarial neural networks to learn a generative model of the data and have proven to be very popular and useful in other fields such as image generation. I like the premise of the paper, and indeed I think GANs can be useful for such tasks.

However, I have several major concerns about the attention to details of the problem, presentation and validation of the work here, as well as lack of comparison to existing methods:

1. The authors state that limited observations can lead to sampling bias. However, a GAN learning from such limited observations, combined with the mode collapse can make the situation worse and lead to amplify biases. How do the authors propose to address this?
2. It seems that the workaround authors have to ensure generation of all cell "types" is conditioning on certain cell types. However, I find this whole approach to be oversimplifying and negates some of the usefulness of the model. First, cell do not exist in discrete clusters. While a partitioning algorithm may be useful for describing or summarizing data, cells could exist in a continuum or progression. Thus rather than trying to regenerate clusters it may be better to try to recreate clusters in the data.
3. tSNE is not a very good method for evaluation of results. tSNE tends to arbitrarily spread out data due to the adaptive kernel. It has a distinct "shattering problem." The fact that all variability is squeezed into two dimensions makes it lose structure. It would be better to compare results using some notion of distribution distances.
4. A major problem ignored in this work is that single cell RNA sequencing data is highly sparse and noisy to begin with. Although GANs can denoise the data to some extent, the results should be compared to an imputation method such as MAGIC (van Dijk et al. Cell 2018) or scImpute (Li et al. Nature Communications 2018). Otherwise, perhaps its better to impute first before training the GAN. Either way this issue needs to be examined.
5. The restoration of relationships in data (measured by mutual information or correlation), as shown by MAGIC can be a key use of data generation and this should be checked.
6. The comparisons to previous works are weak. How about comparisons to variational autoencoders? There is a method called SUGAR (published in NIPS 2018) that generates data from a manifold geometry model of data, and explicitly tries to generate in sparse areas by generating in areas within the manifold that are less well sampled. I think this work should be compared in detail to SUGAR. Due to the manifold model used, it can generate continuum data. <https://arxiv.org/abs/1802.04927> (Lindenbaum et al. Arxiv 2018)
7. The motivation for this problem, while abundantly clear to those in the field is meagerly stated.

Reviewer #2 (Remarks to the Author):

CONTRIBUTION SUMMARY

=====

The paper proposes to build a generative model of the per-cell gene expression from single-cell RNA-seq data. The proposed reason for doing so is to generate extra samples for The generative model used is a GAN with fully-connected layers comprising both generator and discriminator.

The paper juxtaposes scGAN with a hand-written and statistically-oriented scRNA-seq simulation tools like Splatter. A key claim is that scGAN learns interdependencies among gene expression counts better than Splatter.

The method is demonstrated on published datasets of per-cell expression levels inferred from scRNA-seq samples. One sample is of 68,579 healthy PBMC cells, another is of 1.3M mouse brain cells.

The paper also

The paper suggests that it is helpful to normalize the training data to total per-cell read counts, to remove the unknown scaling factor influencing the representation of each cell in the sample.

The qualitative measure of success is whether generated single-cell expression levels appear to overlap real single-cell expression levels in a 2D t-SNE embedding.

A key quantitative measure of success is whether a random forest can distinguish between real and generated expression values when represented via 50 principle components.

REVIEW SUMMARY

=====

I suggest major revisions.

I think that for any "generative model for data augmentation" paper, the burden is on the authors to systematically demonstrate that the generative model (GAN generator in this case) is somehow more sample efficient than the classifier it is supposedly helping after the fact (in this case a random forest). In this paper, the crux of it is understanding whether Figure 3A is progress or an artifact of the choice of generator/classifier and the training regimen used (hyperparameters, held out data, etc). In other words, if there is so little data that a classifier (random forest) isn't doing a good job at discriminating minority class vs majority class, then why should we think that a generative model (GAN generator) trained on that same small amount of minority data will do a better job of modeling the minority class than the classifier could with the same data? It all seems a little magical, and that's why it's important to rule out possibilities like "Oh, the random forest was trained in a way that it was allowed to over-fit the small minority samples and that's why there seemed to be a boost when adding samples from the generator, even if the generator isn't particularly good." Most of my major comments are directed towards a little more rigor around this aspect, even though I'm not sure how to address it definitively. I also have suggestions around clarity and possible fairness to Splatter.

I should state that I have a generally positive outlook on building simulations (by hand and by GAN) and leveraging simulated data for ML, so I'm effectively an enthusiast of the larger approach of which this manuscript is an instantiation. I may be biased towards acceptance compared to the average reviewer.

MAJOR COMMENTS

=====

On how RF classification difficulty is evaluated:

The manuscript shows that a random forest struggles to classify real from simulated data when both are represented by the 50 principle components of the $\sim 17,000$ gene expression values (Cell Ranger dimensionality reduction). This is akin to showing how good a face generator is by first doing PCA on a dataset of faces and then showing that a random forest struggles to classify on that lower-dimensional representation. I agree that this result is meaningful in relative terms, i.e. PCA should capture much of what makes the data difficult to separate and that a lower AUC there is suggestive of a better generator. But, in absolute terms, it is hard to say whether a real-vs-fake classification AUC of 0.65 indicates progress in capturing the data distribution over, say, mixture models. I don't have the expertise to dictate what exact baseline is state of the art for capturing single cell RNA-seq, but I think it's appropriate include a mixture model which can easily be sampled from, such as ZINB (zero-inflated negative binomial)

<https://genomebiology.biomedcentral.com/articles/10.1186/s13059-018-1406-4>, even if a tool like Splatter can't infer reasonable parameters from data (see later comments).

If the authors feel strongly otherwise or think that I have misunderstood the point of demonstrating the fidelity of their generator, I will try to understand their response, but the onus is on them to clarify in the manuscript as well. Also, as a quick sanity check, at the end of training, what is the accuracy of the discriminator at classifying real versus generated data? If the AUC of the discriminator is close to 0.5 then the neural network is (surprisingly) worse than the random forest, but if the AUC is high then it suggests that a neural network would have no trouble telling the real from simulated data in its full gene-level representation.

On using generated samples for improving classifiers:

When training the RFs (Figure 3A and Methods, "Classification of cell sub-populations"). First, using the same hyperparameters for the random forest across all subsets of data is inappropriate. One fixed set of hyperparameters will either over-fit the small samples (artificially inflating the benefit of adding samples from scGAN or cscGAN) or will under-fit the full samples. This needs to be addressed.

Second, if class weights were always used to account for class imbalance, why would duplicating some of the training points (RF upsampled) help? For example, shouldn't a weight of 2 on the smaller class build a classifier that's equivalent to duplicating all the training points in that class? Some other confusing choices in this section (70% of other cells for RF downsampled, but 100% of other cells for RF augmented).

On Splatter comparison:

Despite a valiant attempt at explaining why the Splatter scRNA-seq simulator cannot simulate gene-gene dependencies in scRNA-seq data (Methods, "Splatter comparison") I am still confused by the claim that what is depicted in Figure S4 is the most structure that one can model with Splatter. Even if the gene expression values within a single cell type 'group' in splatter are sampled independently, Splatter can be manually configured to simulate a wide range of joint distributions over expression by using many small groups, much as a mixture of Gaussians can approximate complex distributions by adding more components. However, I can believe that Splatter's "phase 1" does a poor job at inferring gene expression parameters that recapitulate a real scRNA-seq dataset, resulting in a subsequent simulation ("phase 2") that fails to capture the data distribution. Either this distinction should be explained in the paper or, if I have misunderstood, then paper should do a better job of ruling out this misunderstanding about what, exactly, is being criticised in Splatter.

On demonstrating higher-order dependencies in the model:

The manuscript shows good match between the marginal distributions of gene expression and good pairwise correlations. In order to demonstrate that there is higher order dependencies (justifying deep learning), I suggest a t-SNE plot similar showing the data (say, in orange), samples from the marginals where $P(\text{gene}_1, \dots, \text{gene}_n) = P(\text{gene}_1) \dots P(\text{gene}_n)$ (in whatever color), and samples from the full model (in blue). I think this should be easy to do: it basically involves training generative models of the marginals $P(\text{gene}_i)$, drawing samples from those marginals. Those samples should appear off-manifold (far from the data) compared to the samples from the full model. The marginal samples might end up similar to Figure S4B (Splatter).

On evaluating cscGAN:

I may have missed this, but I think it's appropriate to use training individual GANs for each cluster as a baseline, and showing that the PCGAN framework improves upon that (if it actually does) via sharing parameters across clusters. Otherwise the use of PCGAN is needless complexity that makes it harder to build upon this work.

On discussion of training time and running time:

Remove all of this. A sentence describing the kind of system sufficient for training would suffice (e.g. "workstation with GPU takes X minutes to train")

On exposition of GANs:

I think the description of GANs in the main text is unhelpful to readers. Readers who are unfamiliar with GANs should just think of them as being a neural network that transforms a standard distribution (say, N-dimensional normal) to a complex distribution (say, 17,000-dimensional gene expression levels). The generator/discriminator dynamics are beside the point.

On difficulty of evaluating GANs:

Citing [12] for the difficulty of evaluating generative models isn't acceptable to me. Citing Lucic et al 2017 (<https://arxiv.org/abs/1711.10337>) for that discussion is much better.

On using PCA for scRNA-seq data:

Rather than PCA, consider ZIFA (zero-inflated factor analysis) which well-cited, and has a Python library available.

<https://genomebiology.biomedcentral.com/articles/10.1186/s13059-015-0805-z>

(Strongly recommended, though understandable if re-doing experiments and figures is too much to ask at this point. Maybe cite it as possible improvement.)

MINOR COMMENTS (Using page #s from supp DOC file, not PDF.)

=====

Citing Shrivastava et al 2017 as an example of data augmentation with GANs is great, but there's a lot more going on in that paper (refinement of a hand-written forward simulation, GAN stabilization tricks, etc). I think Mariani et al 2018 "data augmentation with balancing GAN" (<https://arxiv.org/abs/1803.09655>) is a very recent preprint, but it seems more directly in the spirit of the submitted manuscript, especially for the cscGAN case, i.e. data augmentation of unbalanced data sets to train classifiers.

p.5 The precise representation of scRNA-seq data you're working with (genes x cells read count matrix) is not described for quite a long time, and lack of concreteness leads to confusion and anxiety while reading the interim text.

a) "in silico generation has seen success in computer vision when used for data augmentation"

b) "Current methods of choice for photo-realistic image generation rely on deep learning-based GANs and VAEs"

Yes, data augmentation schemes require knowledge of invariances in the data (a).

Yes, GANs etc are state of the art for realistic image generation. (b).

But I don't think it directly follows that (b) justifies using GANs for the (a). That needs to be demonstrated, which of course is what this paper is about, but I suggested tweaking the language to make it clear that this is a hypothesis, rather than obviously going to work.

The text isn't really consistent with how much familiarity the reader is presumed to have with GANs or with how deep nets are trained (references how many 'epochs' for training'). Even readers familiar with GANs or scRNA-seq data will know whether "1,500 epochs" is 'only' a little or is a lot or why that would even matter.

p.4: Avoid using "significance" of "statistical analysis." The reader will assume that, later on in the manuscript, samples will be generated to artificially shrink p-values in statistical tests of significance, which would be a completely bogus application of a generative model.

Figure S1 I find it unlikely that a serious attempt at hyperparameter search over number of layers and sizes of those layers would select for steadily increasing/decreasing powers of two and symmetry between generator and discriminator. It's fine, don't need to change anything, just suggestive of intuition-based search so a little old-fashioned.

Why was Louvain clustering necessary to use rather than the more standard k-means workflow? Was the cscGAN is sensitive to that choice, or were the clusters merely more aesthetically pleasing in the figures?

p.3: I'm unsure what is meant by "limited numbers of observations can lead to sampling bias" and how does that "reduce the reproducibility of experimental results"? This sentence seems to be stating something obvious but is written a little too ambiguously.

p.3: Unsure what "classically realistic data modeling" is.

STYLE COMMENTS / TYPOS

=====

Some strange phrasing or word choice, e.g. "unfavourable setting" instead of "unfavourable

situation" or describing priors on invariances as "laws." I hope this gets copy-edited.

p.12: the way (i) is part of a sentence and (ii) is a new sentence is inconsistent.

p.12: the the

p.13: trest

p.23: layrs

p.23: don't -> do not

p.24: missing "W" in "where W and B are its weights and biases"

p.31: descirbed

p.32 poisson -> Poisson

p.33 non-sensical

We would like to thank the reviewers for their careful reading and excellent comments and suggestions. We have addressed all of them, point by point, and think that the revised manuscript is scientifically much stronger.

In this rebuttal, all reviewer comments are listed in black font, while our answers are listed in blue (text changes in black).

Since the revised document is heavily modified, including a novel results section and many novel figures and tables, we did not keep track of changes in the manuscript text itself. The very detailed rebuttal, however, should help the reviewers and the editor to assess the quality and consistency of the manuscript changes.

Lastly, we wish the editor and reviewers Merry Christmas and a Happy New Year.

The authors.

Reviewer 1:

This paper proposes augmenting or generating single-cell RNA sequencing data using a conditional GAN. GANs are neural networks that use two adversarial neural networks to learn a generative model of the data and have proven to be very popular and useful in other fields such as image generation. I like the premise of the paper, and indeed I think GANs can be useful for such tasks.

However, I have several major concerns about the attention to details of the problem, presentation and validation of the work here, as well as lack of comparison to existing methods.

MAJOR COMMENTS

1. The authors state that limited observations can lead to sampling bias. However, a GAN learning from such limited observations, combined with the mode collapse can make the situation worse and lead to amplify biases. How do the authors propose to address this?

This is a very good comment, sampling bias would indeed be amplified and mode collapse is a serious issue for non-Wasserstein GANs, one that we initially struggled with a lot. While the initial seminal works on GANs were suffering heavily from mode collapse, more recent work, notably the Wasserstein GAN, doesn't (or at least to a much lesser extent). We have not observed any evidence of mode collapse for the Wasserstein scGAN and cscGAN implementations across the tested (hyper-) parameter space. We updated the manuscript in the methods section (scGAN model description) to reflect this important point (changes in bold) 'While original GANs⁴ minimize the so-called Jensen-Shannon divergence, they suffer from known pitfalls making their optimization notoriously difficult to achieve. **For instance, they are known to be prone to "mode collapse", where the generated samples are realistic albeit only representing a fraction of the variety of the samples it was trained on (i.e. only a few but not all modes of the distribution of the real samples is learnt)**. On the other hand, "Wasserstein GANs" (WGANs)^{13,25} use a Wasserstein distance, with compelling theoretical and empirical arguments **and show no evidence of mode collapse.**'

Now to the point of sampling bias, in which the reviewer correctly states that a model trained on data that has an overall sampling bias will learn that bias, and replicate or worsen it when generating data. In this manuscript, however, we have many cells from different cell types and we use this information to augment cell types where we have very limited observations. In this initial manuscript we wrote in the introduction 'Finally, we show how our models can successfully augment sparse datasets to improve the quality and robustness of downstream statistical analyses.', which is a statement that is actually wrong. We did not augment 'sparse

datasets' but we augmented sparse 'cell populations', which is why we reworded this sentence to 'Finally, we show how our models can successfully augment sparse cell populations to improve the quality and robustness of downstream statistical analyses.'. We additionally try to highlight this point in the results 'The underlying hypotheses are two-fold. (i) A few cells of a specific cluster might not represent the cell population of that cluster well (sampling bias), potentially degrading the quality and robustness of downstream analyses. (ii) This degradation might be mitigated by augmenting the rare population with cells generated by the cscGAN.' and the discussion 'Most importantly, we provide compelling evidence that generating in silico scRNA-seq data improves downstream applications, especially when sparse and underrepresented cell populations are augmented by the cscGAN generated cells.'. We apologize for our lack of clarity in the initial manuscript and hope the revised version is sufficiently clear.

2. It seems that the workaround authors have to ensure generation of all cell "types" is conditioning on certain cell types. However, I find this whole approach to be oversimplifying and negates some of the usefulness of the model. First, cell do not exist in discrete clusters. While a partitioning algorithm may be useful for describing or summarizing data, cells could exist in a continuum or progression. Thus rather than trying to regenerate clusters it may be better to try to recreate clusters in the data.

The reviewer raises an extremely important point, cells do most probably not 'exist' in discrete clusters but in a continuum. A model should therefore learn a continuous distribution to reflect this biological property.

Both the scGAN and cscGAN model a continuous distribution (as a continuous transform of a continuous distribution in the latent space), yet they differ in their conditioning. While this might be theoretically deduced from the (c)scGAN architectures (only the parameters of the conditional batch normalization are specific to each cluster, while the immense majority of the parameters are shared across all the clusters) it is also supported by our empirical results. The reviewer is completely correct that we haven't shown any evidence of this in the manuscript, neither discussed this important point. In revised the manuscript we have added evidence that scGAN and cscGAN models learn and generate cells of almost identical quality (showing no 'discretization' effect) and added the sentence 'It is interesting to observe that the cscGAN and scGAN generate cells of very similar quality, as a RF classifier reaches an AUC of 0.61 (MMD of 0.674) to distinguish between cscGAN-generated and real cells and an AUC of 0.65 (MMD of 0.547) for scGAN-generated and real cells (differences not significant).' to the results section.

We also agree with the reviewer that cluster recreation is a viable option, the generation of cells using the scGAN could be followed by a cell clustering approach, which would allow to select specific generated cells of interest. Thus, the statement made in the manuscript that the generation of conditional cells is not possible with the scGAN is only partly true ('...', which is not possible with the scGAN model.') and we have changed it accordingly '...', which is not **directly** possible with the scGAN model.'. In addition, we have changed the sentence 'To generate specific cell types of interest while learning multi-cell type complex data, we developed and evaluated various conditional scGAN architectures.' to 'While specific cell types of interest can be obtained by scGAN cell generation followed by clustering and cell selection, we developed and evaluated various conditional scGAN architectures that can directly generate cell types of interest.'.

We would like to thank the reviewer for these extremely valuable comments and we feel that addressing them has made this manuscript scientifically stronger.

3. tSNE is not a very good method for evaluation of results. tSNE tends to arbitrarily spread out data due to the adaptive kernel. It has a distinct "shattering problem." The fact that all

variability is squeezed into two dimensions makes it lose structure. It would be better to compare results using some notion of distribution distances.

We completely agree with the limitations pinpointed by the reviewer regarding t-SNE, which is why we also considered two other qualitative measures in the original manuscript (gene correlation & classification). However, we also agree with the reviewer that a single qualitative score measuring the distance or divergence of real and generated distributions would help to judge the actual performance of the (c)scGANs. We now provide a Maximum Mean Discrepancy (MMD) (Gretton et al. A Kernel Two-Sample Test, Journal of Machine Learning, 2012) distance between simulated and real empirical distributions and updated our manuscript accordingly. We would like to note in this context, that the MMD provides further evidence for the high fidelity of (c)scGAN-generated cells.

4. A major problem ignored in this work is that single cell RNA sequencing data is highly sparse and noisy to begin with. Although GANs can denoise the data to some extent, the results should be compared to an imputation method such as MAGIC (van Dijk et al. Cell 2018) or scImpute (Li et al. Nature Communications 2018). Otherwise, perhaps its better to impute first before training the GAN. Either way this issue needs to be examined.

Please see our answer to major comment 5.

5. The restoration of relationships in data (measured by mutual information or correlation), as shown by MAGIC can be a key use of data generation and this should be checked.

The reviewer raises the point that single cell RNA-seq data is sparse and noisy and wonders if (c)scGANs generate denoised or imputed cells. This is indeed a highly relevant question, which we address in the revised manuscript. In brief, we now highlight the fact that the (c)scGANs are not meant for denoising nor imputation but for 'realistic' data generation. As suggested by the reviewer, we have compared the gene-gene restoration of (c)scGANs to that of MAGIC, which clearly proves that (c)scGANs do not impute expression information. Gene-gene restoration plots for (c)scGANs trained on MAGIC imputed data faithfully represent imputed values, proving that (c)scGANs can faithfully learn and generate imputed data.

We included these results in the revised manuscript in the results 'The realistic modeling of scRNA-seq data entails that our scGANs do not denoise nor impute gene expression information, while they potentially could. Nevertheless, an scGAN that has been trained on imputed data using MAGIC generates realistic imputed scRNA-seq data (Fig. S7).' and in Fig. S7.

6. The comparisons to previous works are weak. How about comparisons to variational autoencoders? There is a method called SUGAR (published in NIPS 2018) that generates data from a manifold geometry model of data, and explicitly tries to generate in sparse areas by generating in areas within the manifold that are less well sampled. I think this work should be compared in detail to SUGAR. Due to the manifold model used, it can generate continuum data. <https://arxiv.org/abs/1802.04927> (Lindenbaum et al. Arxiv 2018)

This is a great suggestion and we have included a comparison to SUGAR in the updated manuscript. While SUGAR generates cells that give good results in t-SNE visualization and gene-gene correlation, it gives significantly higher classification and MMD scores than the (c)scGAN. In conclusion, SUGAR is great for denoising and in conjunction with MAGIC great for getting denoised and imputed scRNA-seq data. We have included this information in the updated manuscript in the results 'SUGAR, on the other hand, generates cells that overlap

with every cluster of the data it was trained on in t-SNE visualizations and accurately reflects cluster-specific gene dependencies (Fig. S6). SUGAR's MMD (59.45) and AUC (0.98), however, are significantly higher than the MMD (0.87) and AUC (0.65) of the scGAN and the MMD (0.03) and AUC (0.52) of the real data (Table S2a, Fig. S6).

The results from the t-SNE visualization, marker gene correlation, MMD, and classification corroborate that the scGAN generates realistic data from complex distributions, outperforming existing methods for in silico scRNA-seq data generation. The realistic modeling of scRNA-seq data entails that our scGANs do not denoise nor impute gene expression information, while they potentially could. Nevertheless, an scGAN that has been trained on imputed data using MAGIC generates realistic imputed scRNA-seq data (Fig. S7).', and Fig. S6.

In addition, we would like to note that we have tested several published scRNA-seq simulation methods, including but not limited to Splatter (Zappia et al. 2017), Lun 2 (Lun et al. 2016) or ZINB-WaVE (Risso et al. 2018) (suggested by reviewer 2). All of them, except for SUGAR, model the gene-cell matrix but generate "virtual genes" that have no one-to-one correspondence with real genes (as explained in the "Splatter comparison" part of our Methods section). As such, they simulate cells that cannot be compared to real cells in tasks such as classification or data augmentation, as we use to evaluate the (c)scGAN.

To our knowledge, the available implementations of Variational Auto-Encoders for single cell data were trained and proposed to find better dimensionality reduction to be used in the downstream analysis and there is no available implementation meant (or allows) to actually simulate scRNA-seq data. It is then unfortunately impossible to use them to simulate comparable cells to compare to scGAN.

In conclusion, we would have loved to compare the (c)scGANs to more simulation algorithms, but could not find any functional implementations. We are therefore very grateful to the reviewer for suggesting SUGAR.

7. The motivation for this problem, while abundantly clear to those in the field is meagerly stated.

We apologize that the first version of the manuscript did not provide a good motivation for our research. In consequence, have reworked large parts of the introduction to provide a clear motivation for (i) problems with small sample sizes, (ii) the description of what a GAN 'does/is', and (iii) the rationale why GANs could be good at generating cells. In detail, we have reworded the sample bias section from 'Thus, limited numbers of observations can lead to sampling bias that could reduce the reproducibility of experimental results, a well-known problem in biomedicine.' to 'Thus, a small sample size might not reflect the population well, an imbalance that can decrease the reproducibility of experimental results.' and added a reference to a review article that explains general problems with small sample sizes in biomedical research (Button et al., 'Power failure: why small sample size undermines the reliability of neuroscience', Nature Rev. Neuroscience, 2013).

In addition, we have changed the description of GANs from 'GANs involve a generator that learns to output realistic in silico generated samples, and a critic that learns to spot the differences between real samples and generated ones (Fig. S1). An 'adversarial' training procedure allows for those two Neural Networks to compete against each other in a mutually beneficial way..' to one that is more accessible to the broad readership of Nature Communications 'GANs involve a generator that outputs realistic in silico generated samples. This is achieved with a neural network that learns to transform a simple, low-dimensional distribution into the true distribution of the samples it is trained on (Fig. S1)'. We have also included another recent reference discussing data augmentation using GANs (Mariani et al. 2018, <https://arxiv.org/abs/1803.09655>).

Importantly, we have also updated the section where we describe why we investigate GANs for biomedical (scRNA-seq) data augmentation 'While data augmentation has been a recent success story in various fields of computer science, the development and usage of GANs and VAEs for omics data augmentation has yet to be investigated. As a proof of concept that realistic in silico generation could potentially be applied to biomedical omics data, we focus on the generation of single cell RNA (scRNA) sequencing data using GANs.'

We hope that these text changes and the addition of references provide a better research motivation, even for non-specialists, and thank the reviewer for this valuable comment.

Reviewer #2

The paper proposes to build a generative model of the per-cell gene expression from single-cell RNA-seq data. The generative model used is a GAN with fully-connected layers comprising both generator and discriminator.

The paper juxtaposes scGAN with a hand-written and statistically-oriented scRNA-seq simulation tools like Splatter. A key claim is that scGAN learns interdependencies among gene expression counts better than Splatter.

The method is demonstrated on published datasets of per-cell expression levels inferred from scRNA-seq samples. One sample is of 68,579 healthy PBMC cells, another is of 1.3M mouse brain cells.

The paper suggests that it is helpful to normalize the training data to total per-cell read counts, to remove the unknown scaling factor influencing the representation of each cell in the sample.

The qualitative measure of success is whether generated single-cell expression levels appear to overlap real single-cell expression levels in a 2D t-SNE embedding.

A key quantitative measure of success is whether a random forest can distinguish between real and generated expression values when represented via 50 principle components.

REVIEW SUMMARY

=====

I suggest major revisions.

I think that for any "generative model for data augmentation" paper, the burden is on the authors to systematically demonstrate that the generative model (GAN generator in this case) is somehow more sample efficient than the classifier it is supposedly helping after the fact (in this case a random forest).

In this paper, the crux of it is understanding whether Figure 3A is progress or an artifact of the choice of generator/classifier and the training regimen used (hyperparameters, held out data, etc).

In other words, if there is so little data that a classifier (random forest) isn't doing a good job at discriminating minority class vs majority class, then why should we think that a generative model (GAN generator) trained on that same small amount of minority data will do a better job of modeling the minority class than the classifier could with the same data? It all seems a little magical, and that's why it's important to rule out possibilities like "Oh, the random forest was trained in a way that it was allowed to over-fitt the small minority samples and that's why there seemed to be a boost when adding samples from the generator, even if the generator isn't particularly good."

Most of my major comments are directed towards a little more rigor around this aspect, even though I'm not sure how to address it definitively. I also have suggestions around clarity and possible fairness to Splatter.

I should state that I have a generally positive outlook on building simulations (by hand and by GAN) and leveraging simulated data for ML, so I'm effectively an enthusiast of the larger approach of which

this manuscript is an instantiation. I may be biased towards acceptance compared to the average reviewer.

MAJOR COMMENTS

1. On how RF classification difficulty is evaluated:

i) The manuscript shows that a random forest struggles to classify real from simulated data when both are represented by the 50 principle components of the ~17,000 gene expression values (Cell Ranger dimensionality reduction). This is akin to showing how good a face generator is by first doing PCA on a dataset of faces and then showing that a random forest struggles to classify on that lower-dimensional representation. I agree that this result is meaningful in relative terms, i.e. PCA should capture much of what makes the data difficult to separate and that a lower AUC there is suggestive of a better generator. But, in absolute terms, it is hard to say whether a real-vs-fake classification AUC of 0.65 indicates progress in capturing the data distribution over, say, mixture models. I don't have the expertise to dictate what exact baseline is state of the art for capturing single cell RNA-seq, but I think it's appropriate include a mixture model which can easily be sampled from, such as ZINB (zero-inflated negative binomial) <https://genomebiology.biomedcentral.com/articles/10.1186/s13059-018-1406-4>, even if a tool like Splatter can't infer reasonable parameters from data (see later comments). If the authors feel strongly otherwise or think that I have misunderstood the point of demonstrating the fidelity of their generator, I will try to understand their response, but the onus is on them to clarify in the manuscript as well.

ii) Also, as a quick sanity check, at the end of training, what is the accuracy of the discriminator at classifying real versus generated data? If the AUC of the discriminator is close to 0.5 then the neural network is (surprisingly) worse than the random forest, but if the AUC is high then it suggests that a neural network would have no trouble telling the real from simulated data in its full gene-level representation.

We will start by answering sub-comment (ii) 'what is the AUC of the discriminator of the network'. This is indeed an interesting question for classical GANs, which rely on a discriminator (classifier) as a proxy to estimate the Jensen-Shannon divergence between the distributions of the real and of the generated samples. Wasserstein GANs, however, have a critic instead of a discriminator, which serves as a proxy to estimate the Wasserstein distance. They do not have a classifier. As a consequence, it is unfortunately impossible for us to assess the fidelity of the (c)scGAN using its AUC.

Now to sub-comment (i), the comparison to existing scRNA-seq simulation tools and the 'meaning' of an AUC of '0.65'. In a nutshell, and quite disappointingly, we only found a single published tool, namely SUGAR (please see comments of reviewer 1) that was able to model realistic cells. While SUGAR generates cells that give good results in t-SNE visualization and gene-gene correlation, it gives significantly higher classification and MMD scores than the scGAN. We have included this information in the updated manuscript in the main text 'Finally, we compared the results of our scGAN model to two state-of-the-art scRNA-seq simulations tools, Splatter and SUGAR (see Methods for details). While Splatter models some marginal distribution of the read counts well (Fig. S5), it struggles to learn the joint distribution of these counts, as observed in t-SNE visualizations with one homogenous cluster instead of the different subpopulations of cells of the real data, a lack of cluster-specific gene dependencies, and a high MMD score (129.52) (Table S2a, Fig. S4). SUGAR, on the other hand, generates cells that overlap with every cluster of the data it was trained on in t-SNE visualizations and

accurately reflects cluster-specific gene dependencies (Fig. S6). SUGAR's MMD (59.45) and AUC (0.98), however, are significantly higher than the MMD (0.87) and AUC (0.65) of the scGAN and the MMD (0.03) and AUC (0.52) of the real data (Table S2a, Fig. S6).', Table S2, and Fig. S6.

All other tools we tested, including but not limited to ZINB-WaVE (suggested by the reviewer) and Lun2, suffer from some of the same limitations as Splatter. They model the matrix as a whole and not each gene specifically. The data produced by those methods give expression levels of "virtual genes", that have no one-to-one correspondence with real genes. While comparing t-SNE plots of cells generated by those models makes sense, using a classifier to discriminate against real cells does not since there is no virtual to real gene correspondence.

We claim, however, that the AUC obtained (where it applies) gives some measure of the realism of the cells and that they can be compared with one another to assess which model produces the more realistic cells. In this context we would like to highlight that classification is not the only quantitative measure of 'realism' we have used in the manuscript. Realistic generation of cells was also measured by gene-gene correlation and in the updated manuscript by MMD, all of which support the notion that (c)scGANs generate high fidelity cells, better than any existing tool we know of (which is additionally supported by our augmentation results).

2. On using generated samples for improving classifiers: When training the RFs (Figure 3A and Methods, "Classification of cell sub-populations").

First, using the same hyperparameters for the random forest across all subsets of data is inappropriate. One fixed set of hyperparameters will either over-fit the small samples (artificially inflating the benefit of adding samples from scGAN or cscGAN) or will under-fit the full samples. This needs to be addressed.

Second, if class weights were always used to account for class imbalance, why would duplicating some of the training points (RF upsampled) help? For example, shouldn't a weight of 2 on the smaller class build a classifier that's equivalent to duplicating all the training points in that class? Some other confusing choices in this section (70% of other cells for RF downsampled, but 100% of other cells for RF augmented).

i) We agree with the reviewer that optimizing the RF hyperparameters could increase the overall RF classification performance. While we could argue that we selected RFs and their default hyper-parameter settings because of their stability across hyperparameter space and robustness against overfitting via sample and feature bootstrapping, we do agree that hyper-parameter optimization would increase the validity of our results. We have included the results of RF hyper-parameter optimization across the number of trees and features for scRNA-seq data augmentation in the revised manuscript. In a nutshell, while the overall performance of the optimized RFs increases, RFs trained with augmented data show almost no performance loss when few real samples were available. This does not hold for upsampled or balanced approaches, replicating the initial results. We have updated the revised manuscript in the results 'In line with this assumption, the blue curves in Figure 3A and Fig. S14 show that augmenting the cluster 2 population with cluster 2 cells generated by the cscGAN almost completely mitigates the effect of the downsampling (F1 score of 0.93 obtained for a downsampling rate of 0.5%). **We obtained similar results with RFs that have been optimized for the number of trees and features per tree (Fig. S14D), showing that augmentation robustly increases classification performance across RF hyper-parameter space.**' and Fig S14.

We would like to thank the reviewer for the very helpful suggestion and believe that the hyper-parameter optimization has strengthened the results.

ii) This is a very good observation and we have to admit that we were equally puzzled when realizing the difference in performance for upsampled (RF upsampled) and downsampled (RF downsampled) training.

In a nutshell, using different class weights or upsampling are similar but not equivalent methods and yield different results. We chose these to approaches based on a literature survey of which methods might yield best results. A good overview of several approaches to deal with class imbalances is given by Chen, Liaw, and Breimann (<https://statistics.berkeley.edu/sites/default/files/tech-reports/666.pdf>).

The motivation for both experiments is that MLs, especially RFs, are very sensitive to class imbalances, resulting in models that 'ignore' small clusters.

We added a more detailed description and explanation of the 'upsampling' and 'balancing' approaches to the methods section of the revised manuscript.

- 3. On Splatter comparison:** Despite a valiant attempt at explaining why the Splatter scRNA-seq simulator cannot simulate gene-gene dependencies in scRNA-seq data (Methods, "Splatter comparison") I am still confused by the claim that what is depicted in Figure S4 is the most structure that one can model with Splatter. Even if the gene expression values within a single cell type 'group' in splatter are sampled independently, Splatter can be manually configured to simulate a wide range of joint distributions over expression by using many small groups, much as a mixture of Gaussians can approximate complex distributions by adding more components. However, I can believe that Splatter's "phase 1" does a poor job at inferring gene expression parameters that recapitulate a real scRNA-seq dataset, resulting in a subsequent simulation ("phase 2") that fails to capture the data distribution. Either this distinction should be explained in the paper or, if I have misunderstood, then paper should do a better job of ruling out this misunderstanding about what, exactly, is being criticised in Splatter.

We agree that we were not very specific in our explanation for why Splatter fails to capture more dependencies between genes. As the reviewer pinpointed, their graphical model is indeed expressive enough to capture complex relationship between genes. And we also agree that it is very likely that it is the inference algorithm ("phase 1") that is failing at capturing the gene-gene relations in the PBMC dataset, even though we don't have further evidence that it is the case. We updated the manuscript accordingly in the Methods section 'It reveals that while those most highly variable genes in the real cells or those generated by the scGAN exhibit some strong correlations, highly variable genes are mostly independent from each other in the cells generated by Splatter. These results are surprising given that the graphical model used in Splatter is expressive enough to accommodate for complex dependencies between genes. It is likely that it is the inference algorithm that is failing at capturing the gene-gene dependencies in the PBMC dataset, while a manual selection of the parameters of Splatter can allow to simulate cells with some gene-gene dependencies.'

- 4. On demonstrating higher-order dependencies in the model:** The manuscript shows good match between the marginal distributions of gene expression and good pairwise correlations. In order to demonstrate that there is higher order dependencies (justifying deep learning), I suggest a t-SNE plot similar showing the data (say, in orange), samples from the marginals where $P(\text{gene}_1, \dots, \text{gene}_n) = P(\text{gene}_1) \dots P(\text{gene}_n)$ (in whatever color), and samples from the full model (in blue). I think this should be easy to do: it basically involves training generative models of the marginals $P(\text{gene}_i)$, drawing samples from those marginals. Those samples should appear off-manifold (far from the data) compared to the samples from the full model. The marginal samples might end up similar to Figure S4B (Splatter).

The reviewer raises the question if the model indeed learns higher-order dependencies, which is an interesting theoretical question. In the original manuscript we have only hypothesized that the model might be able to transfer knowledge from many learnt cells to model sparse cell populations realistically in the results 'The base assumption is that the cscGAN might be able to learn good representations for small clusters by using gene expression and correlation information from the whole dataset.', 'These results strongly suggest that the cluster-specific expression and gene dependencies are learnt by the cscGAN, even when very few cells are available.', and the discussion 'It may be surprising or even suspicious that our cscGAN is able to learn to generate cells coming from very small sub-populations (e.g. 16 cells) so well. We speculate that although cells from a specific type may have very specific functions, or exhibit highly singular patterns in the expression of several marker genes, they also share a lot of similarities with the cells from other types, especially with those that share common precursors. In other words, the cscGAN is not only learning the expression patterns of a specific sub-population from the (potentially very few) cells of that population, but also from the (potentially very numerous) cells from other populations.'

While interesting in nature, it is very difficult to prove that the model learns high-order dependencies. We also would like to state that the definition of what constitutes 'higher-order dependencies' might be ambiguous. Nevertheless, we have taken three different approaches to provide evidence for learnt 'higher-order dependencies' and the transfer of knowledge between different cell clusters. Throughout the manuscript, however, we avoid the usage of the term higher-order dependencies, since we obtained hints but no final proof.

In this context we would like to note that while we cannot make any definite statements about higher-order dependencies, we prove that cscGANs model scRNA-seq data with higher fidelity than any other published tool we know of.

i) We would first like to specifically address the suggestion of the reviewer by removing gene-dependencies from the data. In technical terms we aim to remove any conditional information by ensuring $P(\text{gene}_1, \dots, \text{gene}_n) = P(\text{gene}_1) \dots P(\text{gene}_n)$. To this end, we generated a gene expression matrix with our scGAN and for each gene of the matrix, proceeded to independently and randomly shuffle the expression levels of the different cells. The resulting "shuffled" expression matrix thus preserves the marginals $P(\text{gene}_i)$ for each gene, but drops any dependency between the genes, due to the shuffling. As expected by the reviewer, the resulting samples appear completely off-manifold compared to the samples for the full model, as displayed in the provided t-SNE plot.

A Real vs. Shuffled

B

C

Fig. R1: *scGANs trained on data with shuffled gene expression fail to learn gene dependencies.* Real, unshuffled data is labelled with 'Real', scGAN-generated data based on a shuffled expression matrix as 'Shuffled'. A: t-SNE representations, B: Gene correlation heatmap, and C: ROC of real and shuffled scGAN data.

We would like to highlight that we have not included these results in the revised manuscript but would do so if the reviewer deems them relevant enough.

ii) A hallmark of transcriptional networks is the causal relation between Transcription Factors (TF) and their target genes. TFs define gene regulatory networks that in turn define the cellular states, knowledge that could be learnt by the model. A base unit in a gene regulatory network is termed a regulon, which constitutes a TF and the target genes it regulates. The base assumption of this analysis is that if the (c)scGANs learn higher-order gene dependencies they should be able to learn cell type- and cluster-specific regulons. To evaluate whether scGAN is able to simulate and preserve higher-order gene dependencies, we used the SCENIC package to construct gene modules. In a nutshell, the results confirm that our (c)scGAN indeed learns higher-order dependencies beyond pairwise correlations. Please see the added section in the Methods part of our manuscript for more details. Further, we wanted to evaluate whether scGAN is able to simulate and preserve more complex gene dependencies. To do so, we followed the computational approach of SCENIC to find the regulons in the Ziesel RNA-seq dataset. This dataset was used by the authors of SCENIC to show cross-species Dlx1 regulon activity.

We selected the top 50 target genes with highest weight for each transcription factor (TF) to reduce the computational time and to avoid many arbitrary thresholds. Next, the SCENIC

algorithm applies the motif enrichment framework RcisTarget, which finds significant over-represented TF-binding motifs on a set of genes. Modules with enriched TF-binding motifs were kept and defined as active regulons.

Afterwards, we trained an scGAN model on the Ziesel dataset and used it to generate 10,000 library size normalized cells. The Dlx1 regulon was then found in the real dataset (realDlx1) as well as in the generated one (genDlx1). In addition, we used AUCell to calculate the regulon binarized activity of realDlx1 in the cells of the generated dataset and, vice versa. Reciprocal activity of realDlx1 and genDlx1 regulons are visualised using t-SNE on real and generated data (Fig. S4). We also added in the Results section : 'Furthermore, the scGAN is able to model inter gene dependencies and correlations, which are a hallmark of biological gene-regulatory networks. To prove this point we computed the correlation and distribution of the counts of cluster-specific marker genes (Fig. 1B) and 100 highly variable genes between generated and real cells (Fig. S4). We then used SCENIC to understand if scGANs learn regulons, the functional units of gene regulatory networks consisting of a transcription factor and its downstream regulated genes. scGANs trained on all cell clusters of the Zeisel dataset (see Methods) faithfully represent regulons of real test cells, as exemplified for the Dlx1 regulon in Fig. S4C, suggesting that scGANs learn dependencies between genes beyond pairwise correlations.'

iii) Third and last, we argue that the strongest proof of learning of potentially higher-order dependencies would be if the (c)scGANs could actually transfer knowledge from large clusters to smaller, sparse ones. A detailed motivation and results can be found in the answer to the reviewer's main comment 5. In brief, in the revised manuscript we have trained scGANs on various sizes of cluster 2 (c2 scGANs) and compared the fidelity of the generated cells to that of cscGANs trained on the same cluster 2 cells combined with all other cell clusters. We used a RF classifier and MMD scores to quantify the fidelity by comparing cluster 2 cells generated by the c2 scGAN to real test cluster 2 cells and clusters 2 cells generated by the cscGAN to real test cluster 2 cells. While the c2 scGANs generate more realistic cells than the cscGAN when large numbers of training samples are available, they fail to learn realistic representations when only few training cells are available. The cscGAN, however, generates high fidelity cluster 2 cells, even when as little as 16 cluster 2 cells are available.

These results provide strong evidence for the hypothesis that the (c)scGANs transfer knowledge from large cell clusters to sparse cell clusters, which might or might not be information of higher order.

We have added the results of parts (ii) and (iii) to the revised manuscript as a new section with the title 'cscGANs learn and translate gene regulatory syntax'.

5. **On evaluating cscGAN:** I may have missed this, but I think it's appropriate to use training individual GANs for each cluster as a baseline, and showing that the PCGAN framework improves upon that (if it actually does) via sharing parameters across clusters. Otherwise the use of PCGAN is needless complexity that makes it harder to build upon this work.

This is an excellent suggestion, which we have also addressed in part (iii) of the answer to main comment 4 of the reviewer.

In the revised manuscript, we have trained GANs on various sizes of cluster 2 (c2 GANs) and compared the fidelity of the generated cells to that of cscGANs trained on the same cluster 2 cells and all other cell clusters. We used a RF classifier and MMD scores to quantify the fidelity by comparing cluster 2 cells generated by the c2 GAN to real test cluster 2 cells and clusters 2 cells generated by the cscGAN to real test cluster 2 cells. While the c2 GANs generate more realistic cells than the cscGAN when large numbers of training samples are available, they fail to learn realistic representations when only few training cells are available. The cscGAN,

however, generates high fidelity cluster 2 cells, even when as little as 8 cluster 2 cells are available.

These results provide strong evidence for parameter sharing across clusters and for the hypothesis that the (c)scGANs transfer knowledge from large cell clusters to sparse cell clusters, which might or might not be information of higher order. As already mentioned in the answer to main comment 4, we have added these insights to the revised manuscript as a new section with the title 'cscGANs learn and translate gene regulatory syntax'.

In the end we would like to highlight that while these observations seem quite 'magical', they are also not completely surprising, as a good latent space representation of the learnt clusters should allow for the 'realistic' generation of even 'unseen' events (0 training cells of a cluster of interest), a topic of immense potential but beyond the scope of this manuscript.

- 6. On discussion of training time and running time:** Remove all of this. A sentence describing the kind of system sufficient for training would suffice (e.g. "workstation with GPU takes X minutes to train")

We agree, a discussion on runtime and 'epochs' is too technical for the main text. The important part of this section is, however, that scGANs generate realistic cells across tissues, organisms, data size and complexity, demonstrating the robustness of the method. We reworded the section title to 'Realistic modeling of cells across tissues, organisms, and data size' and rewrote the complete section 'We next wanted to assess how faithful the scGAN learns very large, more complex data of different tissues and organisms. We therefore trained the scGAN on the currently largest published scRNA-seq data set consisting of 1.3 million mouse brain cells and measured both the time and performance of the model with respect to the number of cells used (Table S1, Fig. S9). Qualitative assessment using t-SNE visualization shows that the scGAN generates cells that represent every cluster of the data it was trained on and the expression patterns of marker genes are accurately learnt (Fig. S9).

Interestingly, our results indicate that the runtime to accurately train an scGAN could be scaling sub-linearly with respect to the number of training cells (see Methods for details). In this context it should be noted that the actual time required to train an scGAN depends on the data size and complexity and on the computer architecture used, necessitating at least one high-performance GPU card.

Our results demonstrate that the scGAN performs consistently well on scRNA-seq datasets from different organisms, tissues, and with varying complexity and size, learning realistic representations of millions of cells.' All runtime details are shifted to the supplements, including a brief description of the computer architecture the models were trained on.

- 7. On exposition of GANs:** I think the description of GANs in the main text is unhelpful to readers. Readers who are unfamiliar with GANs should just think of them as being a neural network that transforms a standard distribution (say, N-dimensional normal) to a complex distribution (say, 17,000-dimensional gene expression levels). The generator/discriminator dynamics are beside the point.

We agree and changed the manuscript accordingly 'GANs involve a generator that outputs realistic in silico generated samples. This is achieved with a neural network that learns to transform a simple, low-dimensional distribution into the true distribution of the samples it is trained on (Fig. S1).'

8. **On difficulty of evaluating GANs:** Citing [12] for the difficulty of evaluating generative models isn't acceptable to me. Citing Lucic et al 2017 (<https://arxiv.org/abs/1711.10337>) for that discussion is much better.

We agree and included the additional reference.

9. **On using PCA for scRNA-seq data:** Rather than PCA, consider ZIFA (zero-inflated factor analysis) which is well-cited, and has a Python library available. <https://genomebiology.biomedcentral.com/articles/10.1186/s13059-015-0805-z> (Strongly recommended, though understandable if re-doing experiments and figures is too much to ask at this point. Maybe cite it as possible improvement.)

Great suggestion. In the updated manuscript we show t-SNE and classification performance of scGAN generated cells processed with ZIFA. 'Of note, the fidelity with which scGANs model scRNA-seq data seems to be stable across several tested dimensionality reduction algorithms (Fig. S8).' and Fig. S8. We also cited ZIFA in the methods section of the revised manuscript. In a nutshell, ZIFA, 50 PCs, and 50 most highly-variable genes give very similar clustering, MMD, and RF classification results, providing strong evidence for the fidelity with which the scGANs learn and represent scRNA-seq data.

MINOR COMMENTS

1. Citing Shrivastava et al 2017 as an example of data augmentation with GANs is great, but there's a lot more going on in that paper (refinement of a hand-written forward simulation, GAN stabilization tricks, etc). I think Mariani et al 2018 "data augmentation with balancing GAN" (<https://arxiv.org/abs/1803.09655>) is a very recent preprint, but it seems more directly in the spirit of the submitted manuscript, especially for the cscGAN case, i.e. data augmentation of unbalanced data sets to train classifiers.

This is an excellent suggestion and we included the reference in the updated manuscript.

2. p.5 The precise representation of scRNA-seq data you're working with (genes x cells read count matrix) is not described for quite a long time, and lack of concreteness leads to confusion and anxiety while reading the interim text.

Thanks to the reviewer for pointing this out. We updated the results section of manuscript for more clarity. Given the great success of GANs in producing photorealistic images, we hypothesize that similar approaches could be used to generate realistic scRNA-seq data (i.e. matrices where each row corresponds to a cell and each column to the expression level of a gene).

3. a) "in silico generation has seen success in computer vision when used for data augmentation"
b) "Current methods of choice for photo-realistic image generation rely on deep learning-based GANs and VAEs"

Yes, data augmentation schemes require knowledge of invariances in the data (a).

Yes, GANs etc are state of the art for realistic image generation. (b).

But I don't think it directly follows that (b) justifies using GANs for the (a). That needs to be demonstrated, which of course is what this paper is about, but I suggested tweaking the language to make it clear that this is a hypothesis, rather than obviously going to work.

This is correct and we have reworded the corresponding section in the introduction accordingly 'While data augmentation has been a recent success story in various fields of computer science, the development and usage of GANs and VAEs for omics data augmentation has yet to be investigated. As a proof of concept that realistic in silico generation could potentially be applied to biomedical omics data, we focus on the generation of single cell RNA (scRNA) sequencing data using GANs.'. We hope that we have addressed this comment satisfactorily.

4. The text isn't really consistent with how much familiarity the reader is presumed to have with GANs or with how deep nets are trained (references how many 'epochs' for training'). Even readers familiar with GANs or scRNA-seq data will know whether "1,500 epochs" is 'only' a little or is a lot or why that would even matter.

This is correct and we have rewritten the section on 'epochs' and 'training', as detailed in the answer to major comment 6.

5. p.4: Avoid using "significance" of "statistical analysis." The reader will assume that, later on in the manuscript, samples will be generated to artificially shrink p-values in statistical tests of significance, which would be a completely bogus application of a generative model.

This is a very valid point and we have removed the terms 'significance' of 'statistical analysis' from the revised manuscript.

6. Figure S1 I find it unlikely that a serious attempt at hyperparameter search over number of layers and sizes of those layers would select for steadily increasing/decreasing powers of two and symmetry between generator and discriminator. It's fine, don't need to change anything, just suggestive of intuition-based search so a little old-fashioned.

We agree with the reviewer's comment. We could not claim an 'optimal' model based on our hyperparameter search strategy. The reason for choosing symmetrical number of neurons per layer between the generator and the critic is influenced by previous successful models by other groups and by computational (run-time) limitations to 'sufficiently' explore hyperparameter space. Choosing "power of two" sizes for batch size and layer size is often advised for computational reasons as hardware (on both CPUs and GPUs) is optimized to handle fast matrix multiplications for those sizes. The question of what is 'sufficiently' is a theoretical one, which is notoriously hard to answer.

7. Why was Louvain clustering necessary to use rather than the more standard k-means workflow? Was the cscGAN is sensitive to that choice, or were the clusters merely more aesthetically pleasing in the figures?

There are many ways to cluster single cell data and we have chosen the one we use on a standard basis in our scRNA-seq analyses. We added the results of a 'standard k-means workflow' to the revised manuscript 'Importantly, the fidelity with which cscGANs model scRNA-seq data seems to be independent of the tested Louvain and K-means clustering algorithms (Fig. S11, Table S4).' Fig. S11 and Table S4, showing that the cscGAN is not biased by the choice of the clustering algorithm.

8. p.3: I'm unsure what is meant by "limited numbers of observations can lead to sampling bias" and how does that "reduce the reproducibility of experimental results"? This sentence seems to be stating something obvious but is written a little too ambiguously.

Yes, we tried to 'state the obvious' but certainly failed to do so. The use of the term 'sampling bias' was inaccurate and we removed it from our manuscript. We have reworded this sentence 'Thus, a small sample size might not reflect the population well, an imbalance that can decrease the reproducibility of experimental results.' and added a reference to a review article (Button et al., 'Power failure: why small sample size undermines the reliability of neuroscience', Nature Rev. Neuroscience, 2013).

9. p.3: Unsure what "classically realistic data modeling" is.

We changed the sentence to 'While classically, data modeling relies on a thorough understanding of the priors on invariances underlying the production of such data, ...' and hope that the reviewer is satisfied.

10. Some strange phrasing or word choice, e.g. "unfavourable setting" instead of "unfavourable situation" or describing priors on invariances as "laws." I hope this gets copy-edited.

Corrected.

11. p.12: the way (i) is part of a sentence and (ii) is a new sentence is inconsistent.

Corrected.

12. p.12: the the

Corrected.

13. p.13: trest

Corrected.

14. p.23: layrs

Corrected.

15. p.23: don't -> do not

Corrected.

16. p.24: missing "W" in "where W and B are its weights and biases"

Corrected.

17. p.31: descirbed

Corrected.

18. p.32 poisson -> Poisson

Corrected.

19. p.33 non-sensical

Corrected from 'It is therefore non-sensical to...' to 'We therefore did not...'

Reviewers' comments:

Reviewer #1 (Remarks to the Author):

While the authors have taken some efforts to address reviewer comments the revisions are still significantly lacking in depth. There are several glaring problems:

First: Neural networks in general can indeed impute missing data. This is based on dimensionality reduction capabilities. I believe with the appropriate architectural choices scGAN could do this as well. See SAUCIE: <https://www.biorxiv.org/content/10.1101/237065v4>

Second: The authors completely misapply SUGAR and therefore miss the entire point of this section. The whole point is to compare scGAN against an algorithm that generates from geometry instead of density. So if you turn off density equalization this is not a "fair comparison." Same with the imputation, this should not be turned off. The entire line of reasoning here is to examine if imputation and density equalization are beneficial for single-cell data and therefore should be incorporated into your model (like by running MAGIC first for example). I would really like these points addressed thoughtfully with a view to solving these problems. Indeed the scGAN method may not have certain properties and these should be discussed even if comparisons fall short.

Reviewer #2 (Remarks to the Author):

I think this paper could reasonably go out. I still find some of the motivation and experiments confusing so I'm still going to provide comments to the authors on this latest manuscript, which I hope they'll take into consideration for the final version.

MINOR COMMENTS / SUGGESTIONS

=====

A central claim of the paper is that the GAN transfers knowledge from well-represented cell types to under-represented cell types. In other words, the few observations from the under-represented cell types effectively fine-tune the output distribution, with most of the complex structure in that distribution learned from the well-represented cell types. Somehow this doesn't come across very clearly in the experiments, even though it is posited as a mechanism towards the end. If there were an experiment that showed a reduction in quality of the minor cell type samples as the major cell type was artificially down-sampled, it would lend more direct evidence to this point. The evidence would be even stronger if downsampling of the "most similar" non-class cells to the class being generated had a stronger degradation effect, i.e. it suggests the GAN is extrapolating/tuning expression patterns based on those adjacent cells.

Each time I read about how the point of this paper is to enhance the "quality" or "robustness" of "downstream analysis" or "evaluation", I cannot imagine what the authors are specifically envisioning. Can the authors better describe the stakeholder that they have in mind, and why he/she would be interested in synthetic samples?

Thoughts on the utility of this paper:

- Using a generative model to determine whether or not more data will help, i.e. if you have collected a small set of gene expression measurements and they have high likelihood under your generative model (or are representative of your GAN samples) then it suggests there's no need to collect more data for that condition / cell type.
- Using a generative model to act as a virtual control in some scenario where it's (for some reason) unreasonable to do the full experiment. For example Fisher et al 2018 (arXiv:1807.03876) generate virtual control arms for clinical trials, so that the number of 'real' controls can be reduced (*if* the real and virtual match well enough). On this subject, the sentence in the conclusion about "building virtual patient cohorts for rare diseases" is unrealistic, I think. Clinical trials for rare diseases tend to have tens of patients; virtual controls are more plausible for common disorders, though.

In my original review, I pointed out that the statement "GANs are useful for scRNA-seq analysis"

doesn't follow from "GANs are useful for data augmentation." The authors revised the phrasing around this but I don't think it fundamentally changes anything. I think that if the paper were more clear on its aims, for example clearly stating that the point of this paper is to perform data augmentation for scRNA-seq data sets so as to, for example, train better classifiers (which the paper does in fact try to do), rather than alluding to vague "downstream analyses," then it would make the connection to data augmentation "at large" more clear.

Based on the "Are GANs created equal?" paper it's not clear that the choice of GAN architecture is so critical in terms of avoiding mode collapse, so I would make less of a deal of this. That paper also shows that the original GAN's non-saturating variant does quite well compared to recent GAN proposals, given the same hyperparameter budget.

Reading the paper again, I think the WGAN description is a sore point, not because it is bad or poorly written, but because it's in that awkward middle where readers who understand WGAN will get nothing from it (there's no "new angle" specific to scRNA-seq data) and readers unfamiliar with WGAN (the vast majority) will not find it sufficiently clear (there's no visual illustration of earth mover). On the one hand it's good to be precise about what's being optimized when WGAN is trained, but I just want to state that the current exposition comes off as trying to impress (duality etc lifted from WGAN paper) rather than trying to illuminate or provide intuition appropriate for the audience.

The new sentence citing Figure S1 ("a neural network that learns to transform a simple, low-dimensional distribution into the true high-dimensional distribution") suggests a depiction of a complex distribution being generated from a simple one, but the figure just shows the scGAN architecture (from original manuscript). I think if the figure depicted some kind of structured / multi-modal distribution being generated from a Gaussian, it would be much more useful to the average reader of this manuscript. Most readers of this manuscript won't know what a GAN does or how it works. Figure 1 of the original GAN paper was excellent in this respect, though it spells out the mechanics of training as well (initialization versus final model) whereas that may be unnecessary in your case, and they depict a very simple distribution whereas you're trying to emphasize a complex non-standard output distribution, i.e. something that appears hard to model with GMMs, hence the need for GAN to model it.

Also, on that same sentence, "true high-dimensional distribution" not sure the emphasis on "low-dimensional" vs "high-dimensional" is inherent to GANs or important here, and the word "true" is a bit strong -- maybe just claim that it learns to "mimick" or "emulate" the complex distribution of the observed data, to the point where machine learning methods cannot distinguish.

On reading Richardson and Weiss 2018 ("On GANs and GMMs", arXiv:1805.12462), the authors may want to consider stating (say, in future work) whether they believe "humble" GMMs might perform well as an alternative to GANs or whether they believe GMMs would inherently fail (and why). After all, most readers who are just trying to go about their jobs would rather use good-ol' GMMs if they worked reasonably well for this application.

For the sentence "Qualitative assessment using t-SNE visualization shows that the scGAN generates cells that represent every cluster of the data it was trained on," if the authors had tried another GAN and found it wanting (in terms of mode collapse), this seems like a good place to include the result showing that WGAN hits more modes than the alternative (assuming fair attempt at tuning the alternative). Again, this would be counter to the "All GANs Created Equal?" paper, but may be true for scRNA-seq still.

We would like to thank the reviewers for their careful reading and excellent comments and suggestions. We have addressed all of them, point by point, and believe that the revised manuscript is scientifically improved.

In this rebuttal, all reviewer comments are listed in black font, while our answers are listed in blue (text changes in black).

Reviewer 1:

While the authors have taken some efforts to address reviewer comments the revisions are still significantly lacking in depth. There are several glaring problems:

We are grateful to the reviewer for the critical comments that are well justified. All comments have been addressed in the revised document.

MAJOR COMMENTS

1. First: Neural networks in general can indeed impute missing data. This is based on dimensionality reduction capabilities. I believe with the appropriate architectural choices scGAN could do this as well. See SAUCIE: <https://www.biorxiv.org/content/10.1101/237065v4>

We agree with the reviewer that GANs with different architectures would most probably impute scRNA-seq data, as highlighted in the first revision of this manuscript (answer to point 5 of reviewer 1):

"The reviewer ... imputed data. We included these results in the revised manuscript in the results 'The realistic modeling of scRNA-seq data entails that our scGANs do not denoise nor impute gene expression information, while they potentially could 24. Nevertheless, an scGAN that has been trained on imputed data using MAGIC generates realistic imputed scRNA-seq data (Fig. S7).' and in Fig. S7."

We want to emphasize again that the focus of this manuscript is on 'realistic' data generation and augmentation for scRNA-seq data, including 'noise' and 'bias'. We also believe that a deeper analysis of data imputation capabilities of scRNA-seq GANs is beyond the scope of this study (please also see reference 24).

2. The authors completely misapply SUGAR and therefore miss the entire point of this section. The whole point is to compare scGAN against an algorithm that generates from geometry instead of density. So if you turn off density equalization this is not a "fair comparison." Same with the imputation, this should not be turned off. The entire line of reasoning here is to examine if imputation and density equalization are beneficial for single-cell data and therefore should be incorporated into . your model (like by running MAGIC first for example). I would really like these points addressed thoughtfully with a view to solving these problems. Indeed the scGAN method may not have certain properties and these should be discussed even if comparisons fall short.

We would like to apologize for misapplying SUGAR. The reviewer is completely correct, SUGAR is geometry based and requires density equalization to perform well. In the first revision we applied SUGAR without density equalization for two main reasons: i) We were and are still not able to run SUGAR with density equalization on the full PBMC scRNA-seq dataset with 68,579 cells and 17,956 genes per cell. ii) We also theorized that density equalization might reduce SUGAR's performance compared to scGANs, which was obviously wrong.

The largest model we could compute with SUGAR and density estimation was for 3,000 cells and 2,000 genes from the PBMC dataset, using 1.3 Terabytes of RAM and computing for over

36 hours. To highlight this point we have measured the time and memory usage of SUGAR for several cell and gene numbers (Fig. 1).

Fig. 1: Runtime (first column) and memory usage (second column) of SUGAR for increasing numbers of genes (first row) and cells (second row). SUGAR used adaptive kernel estimation and density equalization to generate 10,000 cells. The experiments were performed using a Matlab SUGAR implementation on a Dell Power Edge R940 server with 128 x 2.6 GHz Intel Xeon threads and 1.47 Terabytes of RAM. In the end, the largest model we could fit using SUGAR was limited to 3,000 cells and 2,000 genes of the PBMC scRNA-seq dataset, using 1.3 Terabytes of RAM and taking more than 36 hours to complete.

In the revised document, we now compare the performance of scGANs to SUGAR with density equalization and with and without imputation. Both algorithms are trained on the same reduced set of PBMCs (3,000 cells x 2,000 genes) and performance is measured using MMD and classification scores on held-out test cells.

We would like to highlight that this severe reduction in features, from over 17,000 to 2,000 (and observations) goes against our aim to model cells realistically. The sole purpose of reducing the training dataset is to enable a comparison between SUGAR with density equalization and scGANs.

While SUGAR with density equalization creates more realistic cells than SUGAR without density equalization, it has a significantly higher MMD (5.840) and classification (0.89) than the scGAN (2.720, 0.73) (for both values the lower the better). SUGAR with density equalization and imputation generates cells that look 'less realistic', resulting in higher MMD (114.484) and classification (0.99) scores as compared to SUGAR without imputation.

We added the best SUGAR results (density equalization, no imputation) to the revised manuscript while not going into detail on SUGARs runtime and memory limitations. Results for SUGAR with density equalization and imputation are briefly mentioned in the supplements.

Lastly, we would also like to highlight that we performed a hyper-parameter optimization for the scGAN on the reduced-size PBMC dataset, since the original scGAN model was optimized for large datasets (68,579 to over 1 million cells and over 17,000 genes per cell). This resulted in a much smaller scGAN architecture and model training in under an hour on a single high-performance GPU.

In summary, for the realistic generation of scRNA-seq data scGANs seem to outperform current SUGAR implementations both in terms of fidelity and scalability. When data has been imputed before, using for example MAGIC, scGANs can also generate high-fidelity imputed data.

Reviewer 2:

I think this paper could reasonably go out. I still find some of the motivation and experiments confusing so I'm still going to provide comments to the authors on this latest manuscript, which I hope they'll take into consideration for the final version.

We are grateful to the reviewer for the support. All suggestions have been addressed in the revised document.

MINOR COMMENTS / SUGGESTIONS

1. A central claim of the paper is that the GAN transfers knowledge from well-represented cell types to under-represented cell types. In other words, the few observations from the under-represented cell types effectively fine-tune the output distribution, with most of the complex structure in that distribution learned from the well-represented cell types. Somehow this doesn't come across very clearly in the experiments, even though it is posited as a mechanism towards the end. If there were an experiment that showed a reduction in quality of the minor cell type samples as the major cell type was artificially down-sampled, it would lend more direct evidence to this point. The evidence would be even stronger if downsampling of the "most similar" non-class cells to the class being generated had a stronger degradation effect, i.e. it suggests the GAN is extrapolating/tuning expression patterns based on those adjacent cells.

We're sorry to read that the "knowledge transfer" from well-represented to under-represented cell types didn't come across clearly in the results we added in the first revision. Notably, we expected the experiments we conducted and the results reported at the end of the Results section ("cscGAN learns and translates gene regulatory syntax") and in Figure S17 to be eloquent on this issue. In panel B of Figure S17, one can see that while an scGAN trained on a downsampled population tends to only generate cells that only resemble those it saw in the training set (i.e. the red cells are clustered around the black ones). On the other end, the cscGAN, which was trained on the exact same cells in the training is generating much more varied cells that don't only cluster around the black ones but reflect the actual wider variety of this cluster 2 sub-population (light blue dots). This, to us, represents a strong illustration of the knowledge transfer.

2. Each time I read about how the point of this paper is to enhance the "quality" or "robustness" of "downstream analysis" or "evaluation", I cannot imagine what the authors are specifically envisioning. Can the authors better describe the stakeholder that they have in mind, and why he/she would be interested in synthetic samples?

Thoughts on the utility of this paper:

- Using a generative model to determine whether or not more data will help, i.e. if you have collected a small set of gene expression measurements and they have high likelihood under your generative model (or are representative of your GAN samples) then it suggests there's no need to collect more data for that condition / cell type.
- Using a generative model to act as a virtual control in some scenario where it's (for some reason) unreasonable to do the full experiment. For example Fisher et al 2018 (arXiv:1807.03876) generate virtual control arms for clinical trials, so that the number of 'real' controls can be reduced (*if* the real and virtual match well enough). On this subject, the

sentence in the conclusion about "building virtual patient cohorts for rare diseases" is unrealistic, I think. Clinical trials for rare diseases tend to have tens of patients; virtual controls are more plausible for common disorders, though.

Thanks to the reviewer for bringing up this important point and suggesting some interesting potential use cases.

While the potential applications are very wide, we agree that using more specific wording, through the manuscript could help clarify our intentions. In the abstract, we already had used more specific examples of analysis: "Augmenting sparse cell populations with cscGAN generated cells improves downstream analyses such as the detection of marker genes, the robustness and reliability of classifiers, the assessment of novel analysis algorithms, and might reduce the number of animal experiments and costs in consequence.". Please see point 3 of this rebuttal for changes we made in the Introduction and Results sections to match this level of specificity.

In the Discussion section, we again opened up to more varied use-cases: "We hypothesize that data augmentation might be especially useful when dealing with human data, which is notoriously heterogeneous due to genetic and environmental variation. Data generation and augmentation might be most valuable when working with rare diseases or when samples with a specified ethnicity or sex, for example, are simply lacking."

3. In my original review, I pointed out that the statement "GANs are useful for scRNA-seq analysis" doesn't follow from "GANs are useful for data augmentation." The authors revised the phrasing around this but I don't think it fundamentally changes anything. I think that if the paper were more clear on its aims, for example clearly stating that the point of this paper is to perform data augmentation for scRNA-seq data sets so as to, for example, train better classifiers (which the paper does in fact try to do), rather than alluding to vague "downstream analyses," then it would make the connection to data augmentation "at large" more clear.

We completely agree that it is too vague to mention 'downstream analyses' in the introduction and results section of the manuscript. We now highlight in the introduction and results section that we aim to increase the robustness of classifiers by data augmentation. Additional use cases that were not experimentally addressed in this manuscript are restricted to the discussion.

4. Based on the "Are GANs created equal?" paper it's not clear that the choice of GAN architecture is so critical in terms of avoiding mode collapse, so I would make less of a deal of this. That paper also shows that the original GAN's non-saturating variant does quite well compared to recent GAN proposals, given the same hyperparameter budget.

We agree with the reviewer that "Are GANs created equal?" shows that the picture is not as binary as our writing may suggest. However, mode collapse is only one of the examples of what can go wrong when training a GAN. It is our experience that the WGAN-GP, coupled with the use of AMSGrad has made the training more stable and robust to the choice of hyper-parameters. We updated the manuscript to reflect it : "On the other hand, "Wasserstein GANs" (WGANs)^{15,33} use a Wasserstein distance, with compelling theoretical and empirical arguments. **In our hands, WGANs showed no evidence of mode collapse and showed stable and robust training with respect to hyper-parameter optimization.**"

5. Reading the paper again, I think the WGAN description is a sore point, not because it is bad or poorly written, but because it's in that awkward middle where readers who understand WGAN will get nothing from it (there's no "new angle" specific to scRNA-seq data) and readers

unfamiliar with WGAN (the vast majority) will not find it sufficiently clear (there's no visual illustration of earth mover). On the one hand it's good to be precise about what's being optimized when WGAN is trained, but I just want to state that the current exposition comes off as trying to impress (duality etc lifted from WGAN paper) rather than trying to illuminate or provide intuition appropriate for the audience.

We are sorry to read that the current writing of the WGAN description comes off as 'trying to impress'. We agree that the level of details is an 'awkward middle', although we would prefer calling it 'striking a balance'. The main reason was to properly and precisely introduce the objective function optimized by the (c)scGAN while keeping the manuscript self-contained. We hope that it will be enough for the specialist to be sure about what we are precisely doing and enticing enough for the unfamiliar reader to dive into the given references for deeper understanding.

6. The new sentence citing Figure S1 ("a neural network that learns to transform a simple, low-dimensional distribution into the true high-dimensional distribution") suggests a depiction of a complex distribution being generated from a simple one, but the figure just shows the scGAN architecture (from original manuscript). I think if the figure depicted some kind of structured / multi-modal distribution being generated from a Gaussian, it would be much more useful to the average reader of this manuscript. Most readers of this manuscript won't know what a GAN does or how it works. Figure 1 of the original GAN paper was excellent in this respect, though it spells out the mechanics of training as well (initialization versus final model) whereas that may be unnecessary in your case, and they depict a very simple distribution whereas you're trying to emphasize a complex non-standard output distribution, i.e. something that appears hard to model with GMMs, hence the need for GAN to model it.

We added a corresponding graphical representation to the updated Figure S1.

7. Also, on that same sentence, "true high-dimensional distribution" not sure the emphasis on "low-dimensional" vs "high-dimensional" is inherent to GANs or important here, and the word "true" is a bit strong -- maybe just claim that it learns to "mimick" or "emulate" the complex distribution of the observed data, to the point where machine learning methods cannot distinguish.

We changed the sentence in accordance with your suggestion to "This is achieved with a neural network that learns to transform a simple, low-dimensional distribution into a **high-dimensional distribution that is virtually indistinguishable from the real training distribution** (Fig. S1)"

8. On reading Richardson and Weiss 2018 ("On GANs and GMMs", arXiv:1805.12462), the authors may want to consider stating (say, in future work) whether they believe "humble" GMMs might perform well as an alternative to GANs or whether they believe GMMs would inherently fail (and why). After all, most readers who are just trying to go about their jobs would rather use good-ol' GMMs if they worked reasonably well for this application.

This is indeed a very interesting article 'on GANs and GMMs'. While the relevance of such an approach for single cell data remains to be studied, we would like to highlight that the main advantage of these "GMMs" is that they seem to yield a more diverse representation of the original data (at the cost of sharpness). While the dropping of some "rare" types of images is a recurring problem for GANs on images, it does not seem to affect our (c)scGAN (maybe their introduced measure of diversity could be a good way to investigate the issue further). Also,

"good-ol' GMMs" (with full-rank covariance matrices, EM algorithm etc.) definitely do not scale to large-dimensional data. These concerns are addressed in Richardson and Weiss to a certain extent only, as the largest images have size 64x64, which is an order of magnitude smaller than the number of genes in our data.

9. For the sentence "Qualitative assessment using t-SNE visualization shows that the scGAN generates cells that represent every cluster of the data it was trained on," if the authors had tried another GAN and found it wanting (in terms of mode collapse), this seems like a good place to include the result showing that WGAN hits more modes than the alternative (assuming fair attempt at tuning the alternative). Again, this would be counter to the "All GANs Created Equal?" paper, but may be true for scRNA-seq still.

While we did observe mode collapse with classical GANs on MNIST data in our preliminary work, we simply failed at managing to train them on single cell data (the iterates get very unstable and start diverging early in the training). Despite our efforts, we failed at producing anything mildly resembling any single cell sample with a classical GAN. On the other hand, the training with a WGAN required quite some tuning to get to this level of realism but even our first attempts (with the default parameters from the WGAN paper and a vanilla MLP for the critic and generator networks) yielded promising results. As one could always argue that we didn't try enough or used the wrong hyper-parameters for the classic GAN, we elected not to report those results more in detail.

Reviewers' comments:

Reviewer #1 (Remarks to the Author):

While the authors addressed many of the concerns, I still find that a major problem here is that the authors keep focusing on generating rough "clusters" of cells and not on using a sophisticated method like a GAN to generate in all the richness of the cellular phenotypic state space.

Not all cell systems have highly clustered cells. More importantly, it is unclear what the value would be of artificially generating dominant cluster centroid cell types. This would be like repeatedly generating a very canonical Tcell type, perhaps naive cells over and over again. The interesting part of the data, one that is undersampled and NEEDs artificial generation may be a rare regulatory phenotype. In addition to rare populations, indeed there are many systems that are transitional, and even within a cluster there are transitions. The real utility again would be to see if it can generate through the state space.

The whole point of the SUGAR comparison was to check how well scGAN generates transitional and rare populations. The authors repeatedly miss the point here and compare blindly. So for this you would zoom in on one cluster or continuum and see how well different parts of the data space in a continual transitional space would be generated. Are there undersampled populations that are filled in by the scGAN? What is the expected time to fill in the population? Could we conditionally generate undersampled populations? That would be really of value.

I also find the use of MMD to compare between sugar and scGAN to be nonsensical. As is stated in the SUGAR manuscript, SUGAR does not replicate density and EQUALIZES it. So therefore, it makes no sense to use an estimate of probability density to compare. Instead you would check if the cells generated by SUGAR and scGAN are realistic cell types, how far of outliers they are, and if rare populations are well represented.

One metric would be how well scGAN and SUGAR generate sparse parts of the dataspace (that are still well within the dataspace). You could pick these out by performing knn based density estimation on your data graph.

Therefore I suggest in the next revision to include a section that addresses cell continuums and sparse areas of the cellular state space.

Reviewer #2 (Remarks to the Author):

The author responses are mostly reasonable. I didn't have any serious concerns in the last revision (SUGAR is not on my radar), and am still fine with it going out.

MINOR COMMENTS:

On choosing WGAN over GAN: The most recent rebuttal makes it clear that the authors tried training a regular GAN on scRNA-seq but that it failed miserably, but that they shied away from reporting this result because they worry it opens an standard avenue for criticism ("oh, you probably didn't try enough hyperparameters on vanilla GANs"). I can only speak for myself, but as a reader I was much more satisfied to know that the authors tried vanilla-GANs, that they didn't get them to work, that they're open to the possibility that maaaaybe they didn't find the magical hyperparameters to make vanilla-GANs work, but that WGANs were producing reasonable results out of the gate so they focused on WGANs. All of that sounds perfectly reasonable to me, as it's clear to everyone that the purpose of this paper isn't to benchmark GANs vs WGANs or to even give GANs a fair shot, it's to report your experience to the readers so that they can make an

informed decision about what to try. Even if your GAN-vs-WGAN investigations weren't 100% systematic I think this information is valuable and I encourage the authors to mention it, especially given that the paper will likely be accepted (nobody's going to harangue you on Twitter over the GAN hyperparameters).

On describing WGAN: I think the 'precised definition' of the training objective should be moved out of the main text and into the methods section. The authors clearly disagree in their rebuttal, and that's ok, but this is still my editorial advice.

Dimensionality of GAN latent space: It still bugs me that the paper says that a defining characteristic of GANs is to transform a "low" dimensional representation into a "high" a dimensional one. But ok, I suppose that's often the case in practice.

- If you need to cut citations, two t-SNE citations seems redundant. (Also, citation style for van der Maaten is inconsistent.)

- Acronym WGAN is defined twice.

The following lines contain the reviewers' comments (black font) and our point for point rebuttal (blue font, manuscript text in black font):

We suspect that some of the resolved and unresolved discussions arose from a lack of precision on what we mean by “*realistic* generation” and by “*data augmentation*”. In this work “*realistic* generation” is referring to the generation of data that mimics the distribution of the real data. This definition motivates the metrics we used to assess the realism of generated data, MMD score, classification accuracy, and marker gene correlation. Our definition of “*data augmentation*” is the one commonly adopted in the Machine Learning community. It consists of adding to an empirical sample some points which are drawn from the same distribution, so that the distribution of the augmented dataset follows the distribution of the original data. A corollary is that an algorithm that generates “*realistic*” samples (according to the previous definition) should be used for data augmentation.

From these definitions it follows that an algorithm that does not model data or data subsets of interest according to their true underlying probability density function would ‘not produce realistic data’ and would not achieve good MMD, classification, and augmentation results. To avoid any potential confusion, we updated the definitions of ‘realism’ and ‘augmentation’ in the revised manuscript. In the introduction, we added: “In practice, *in silico* generation has seen success in computer vision when used for ‘data augmentation’, whereby *in silico* generated samples are used alongside the original ones to artificially increase the number of observations. **In this manuscript, we focus on augmenting real with newly generated samples, in their original gene space, and whose distribution mimics the original data.**”. Further, at the beginning of the Results section: “Given the great success of GANs in producing photorealistic images, we hypothesize that similar approaches could be used to generate realistic scRNA-seq data (i.e. matrices where each row corresponds to a cell and each column to the expression level of a gene). **In this work ‘realistic’ is referring to the generation of data that mimics the distribution of the real data, without merely replicating them.**”.

Reviewer #1 (Remarks to the Author):

We would like to thank reviewer 1 for the time and effort spent to this manuscript scientifically stronger. While we do not agree with reviewer 1 in all matters, the comments and criticism have not only improved this manuscript but also increased the authors notion of ‘realism’ and ‘partitioning’.

1. While the authors addressed many of the concerns, I still find that a major problem here is that the authors keep focusing on generating rough “clusters” of cells and not on using a sophisticated method like a GAN to generate in all the richness of the cellular phenotypic state space.

The main criticism reviewer 1 is that (point 1) ‘the authors keep focusing on generating rough “clusters” of cells and not on ... to generate all the richness of the cellular phenotypic state space’. More specifically, the reviewer is not happy with selecting partitioning of cells based on clustering algorithms and would rather concentrate on (points 2 & 3 & 5) ‘... transitional and rare populations.’ Especially for rare populations the reviewer suggests to use density estimation to sample from or condition on rare populations, which is at the heart of SUGAR (points 3 & 4) ‘The whole point of the SUGAR comparison was to check how well scGAN generates transitional and rare populations’.

We understand that the reviewer has specific interest in the analysis of a continuum of cells or low density areas. The focus of this manuscript is, however, that (c)scGANs are superior to other tested algorithms in ‘realistic’ scRNA-seq data generation, which can then be used to augment sparse real data. The focus is not on automatically extracting which regions might be the most undersampled or ‘interesting’, which is the case for SUGAR, for example. In a nutshell, (c)scGANs work on arbitrary partitioning choices, including a continuum or low-density regions. To highlight this important point, we added the following sentences to the discussion of the revised manuscript: **“Throughout this manuscript, we solely focused on using cell types as a side information to condition the generation on. It is worth mentioning that any other kind of side information (partitioning of the sample) could equally be used. For instance, a cscGAN could be conditioned and trained on a combination of case and control samples. While many other choices could lead to interesting applications, we leave this avenue of research for future work.”**

The choice to focus on clusters is driven by the fact that most biologists (we know) that work with scRNA-seq data care about cell types, and these occur mostly in clusters. To us it was therefore a natural choice to model and augment cell types (clusters) with a clear biological meaning. We did not and do not want to confound the clear message of the manuscript with notions of other partitions that might, and might not carry clear biological meaning.

This does not imply that (c)scGANs only work on ‘rough clusters of cells’. This is especially true for scGANs for which cell generation is independent of the partitioning algorithms (e.g. density estimation, different clustering algorithms). Generated cells can be extracted *ad libitum* from partitions using any algorithm, as we have highlighted in the answer to the 2nd major concern of reviewer 1 in the first revision. In the manuscript we had added after the first revision: “While specific cell types of interest can be obtained by scGAN cell generation followed by clustering and cell selection, we developed and evaluated various conditional scGAN architectures that can directly generate cell types of interest.”. cscGANs on the other hand are conditioned on the partitions and could theoretically be affected by the choice of algorithm. This is actually the rightful concern of reviewer 2, which we have addressed in the revised manuscript (revision 1 minor comment 7, manuscript Figure S11). It is obvious that we cannot exhaustively test all partitioning algorithms (density, cluster, etc.), which is why we did not extend this section beyond two partitioning algorithms. Again, this would only apply to the cscGAN version, the scGAN is not affected by definition.

In a nutshell, this manuscript is about realistic simulation and augmentation of scRNA-seq data. We chose to partition on cell types (clusters), which is easy to understand and a biologically valuable use case. This manuscript is complex as is and we feel if we now delve into alternate partitioning we would lose the simple message that (c)scGANs are excellent for scRNA-seq data generation and augmentation (please refer to Figure 3).

Lastly, we would like to note that (c)scGANs generate cells that cover ‘all the richness of the cellular phenotypic state space’ and we have not witnessed a single case where a ‘continuum’ or a ‘sparse area’ of cells was not modeled realistically (see Fig. 1A, 2A, 2B)).

2. Not all cell systems have highly clustered cells. More importantly, it is unclear what the value would be of artificially generating dominant cluster centroid cell types. This would be like repeatedly generating a very canonical Tcell type, perhaps naive cells over and over again. The interesting part of the data, one that is undersampled and NEEDS artificial generation may be a rare regulatory phenotype. In addition to rare populations, indeed there are many systems that are transitional, and even within a cluster there

are transitions. The real utility again would be to see if it can generate through the state space.

In addition to our answer to point 1, we would like to note that a cluster has no notion of biology, it just happens to coincide with a 'relatively clean' population of known cell types. These cell types have been defined *a priori*, using various data types and measures. This directly implies that clusters formed by the clustering algorithm can contain arbitrary many cell types, transition states, or just one 'clean' cell type. Many times this is dependent not only on the algorithm but also on the parameter settings. Exactly the same holds true for any other partitioning.

In addition, we would like to highlight that (c)scGANs artificially generate cells resembling real cells in all their variability and not only to the centroids, following our definition of "realistic generation". We would like to highlight that we have addressed this point already in the answer to the 2nd major concern of reviewer 1 in the first revision and in the previous point.

The main message of this manuscript is that (c)scGANs are superior to other tested algorithms in 'realistic' scRNA-seq data generation, which can then be used to augment sparse real data. The focus is not on automatically extracting which regions might be the most undersampled or 'interesting'.

3. The whole point of the SUGAR comparison was to check how well scGAN generates transitional and rare populations. The authors repeatedly miss the point here and compare blindly. So for this you would zoom in on one cluster or continuum and see how well different parts of the data space in a continual transitional space would be generated. Are there undersampled populations that are filled in by the scGAN? What is the expected time to fill in the population? Could we conditionally generate undersampled populations? That would be really of value.

The first part of point 3 is about partitioning, which we have addressed in our answers to points 1 & 2. Here we will focus on answering the three specific questions that reviewer 1 raises.

First of all, human language is ambiguous and we are truly sorry to 'repeatedly miss the point and compare blindly'.

Q1: Are there undersampled populations that are filled in by the scGAN? (c)scGANs model all cells and all 'clusters', sparse and non-sparse ones, and we are not aware that any populations whatsoever are missed, as we stated previously. In part 4 of the results (cscGAN learns and translates gene regulatory syntax) we simulate undersampling by holding out large portions of a specific cluster from the original data and indeed show that cscGAN does fill undersampled populations (see Fig. S17).

Q2: What is the expected time to fill in the population? This would depend on the computer, on the sparsity of the cluster, and on the number of generated cells envisioned. Due to this strong conditional dependence we have taken out runtime estimates from the first version of the manuscript (revision 1, reviewer 2, point 6; manuscript section 'Realistic modeling of cells across tissues, organisms, and data size').

Generating one cell corresponds to one forward pass in a network comprising 3 fully connected layers, which is a matter of milliseconds, and the number of cells generated scale linearly in time. We have not re-included this information in the manuscript but would do so if required.

Q3: Could we conditionally generate undersampled populations? cscGANs can generate realistic cells from partitions of “undersampled” populations. The conditional generation of sparse partitions (in our case a cell cluster) is the focus of parts 2 and 4 of the manuscript. This point is also related to our answer to the 2nd comment of Reviewer 1 in the first revision (‘It seems that the workaround...’).

4. I also find the use of MMD to compare between sugar and scGAN to be nonsensical. As is stated in the SUGAR manuscript, SUGAR does not replicate density and EQUALIZES it. So therefore, it makes no sense to use an estimate of probability density to compare. Instead you would check if the cells generated by SUGAR and scGAN are realistic cell types, how far of outliers they are, and if rare populations are well represented.

One metric would be how well scGAN and SUGAR generate sparse parts of the dataspace (that are still well within the dataspace). You could pick these out by performing knn based density estimation on your data graph.

We agree that given our definition of “realistic generation” from which our performance measures and whole approach derives, it is only natural that an algorithm like SUGAR would fall short if it would equalize distributions. In other words, if SUGAR would sample preferentially from sparse regions in a cluster, MMD scores and also classification results compared to the full cluster distribution would indeed be ‘nonsensical’ (although we would prefer the word ‘incomparable’). To avoid confusion we would like to emphasize this problem, SUGAR with ‘density equalization’ samples preferentially from sparse areas and does therefore not reflect the true distribution of a cluster, which obviously results in higher MMD (and also classification) scores.

Density equalization: The first revision of this manuscript contained a comparison to SUGAR without ‘density equalization’ for reasons given above. In more detail, SUGAR without ‘density equalization’ samples generated cells close to every real training cell by adding ‘noise’ (estimated or fixed covariance, see below). This sampling process replicates the true distribution of the data, which would make MMD and classification a good choice to measure the realism of generated cells.

Reviewer 1 asked us in revision 2, major comment 2 to use SUGAR with ‘density equalization’ and imputation. We heeded to this specific request, which in retrospect was a mistake we corrected in this revised manuscript (see below).

Covariance estimation: While we cannot use ‘density equalization’ due to the above reasons, SUGAR’s ‘covariance estimation’ does not scale to the required number of genes and cells, as we clearly stated in revision 2 (rebuttal to Major Comment 2 of reviewer 1 in the second revision, and in the ‘SUGAR comparison’ methods section of the revised manuscript). In more detail, we argue that a generated cell can be considered realistic (up to our definition) only when it can be produced in its original space (i.e. with counts for the circa 20k genes present in the PBMC dataset, for instance). SUGAR with noise covariance estimation seems to scale quadratically in space with the number of genes and quadratically in space with the number of cells (see Figure 1 in the rebuttal to major comment 2, reviewer 1, second revision). As a result, we were not able to use SUGAR and ‘covariance estimation’ for more than 3k cells (instead of 68k to over 1 million) and 2k genes (instead of almost 20k), requiring 1.3 Tbytes of RAM and 36 hours to generate 3000 new cells from this reduced dataset (on a high-end server with 128 x 2.6GHz Xeon threads and 1.47 Tbytes of RAM). We would also like to remind reviewer 1 that the suggestion to deactivate the noise covariance estimation came directly from the first author of the SUGAR manuscript. To summarize this paragraph, SUGAR with ‘covariance estimation’ does not scale to the necessary number of genes and cells to meet our criterion of realism.

MMD: As stated above, MMD measures distribution similarity and is well suited to compare generated to real cells when SUGAR is used without 'density equalization'. In this context we would like to thank reviewer 1 again for suggesting to use a notion of distribution (reviewer 1, revision 1, main point 3, 'It would be better to compare results using some notion of distribution distances.'). Following this excellent suggestion we used MMD (in addition to tSNE, correlation, and classification) to measure cell realism.

In summary, given our definition of realism and the evaluation criteria used (MMD and classification accuracy), we reverted to the results we obtained in our first revision, where we had deactivated SUGAR's noise covariance estimation and the equalization. This gave rise to the following changes in our revised manuscript: In the first part of the results section: "**SUGAR, on the other hand, generates cells that overlap with every cluster of the data it was trained on in t-SNE visualizations and accurately reflects cluster-specific gene dependencies (Fig. S6). SUGAR's MMD (59.45) and AUC (0.98), however, are significantly higher than the MMD (0.87) and AUC (0.65) of the scGAN and the MMD (0.03) and AUC (0.52) of the real data (Table S2A, Fig. S6)**". We also adapted the 'SUGAR comparison' methods section to reflect the above line of reasoning. Finally we removed the part about the 'Reduced PBMC dataset' in the 'Datasets' section of our methods as it is no longer used.

5. Therefore I suggest in the next revision to include a section that addresses cell continuums and sparse areas of the cellular state space.

While we understand the reviewer's pique interest in a cell continuum and sparse areas of the cellular state space, we feel it might be best addressed in a separate manuscript. We fear that a section on a cell continuum and sparse areas of cellular state space would distract from the main message of realistic scRNA-seq generation and augmentation. We also don't believe a single section would suffice to address the rather complex notions of a cell continuum and sparse areas of cellular state space, both biologically and technically.

Reviewer #2 (Remarks to the Author):

The author responses are mostly reasonable. I didn't have any serious concerns in the last revision (SUGAR is not on my radar), and am still fine with it going out.

We would like to thank reviewer 2 for the legitimate and precise critique. We truly believe it has made this manuscript scientifically stronger.

MINOR COMMENTS:

1. On choosing WGAN over GAN: The most recent rebuttal makes it clear that the authors tried training a regular GAN on scRNA-seq but that it failed miserably, but that they shied away from reporting this result because they worry it opens an standard avenue for criticism ("oh, you probably didn't try enough hyperparameters on vanilla GANs"). I can only speak for myself, but as a reader I was much more satisfied to know that the authors tried vanilla-GANs, that they didn't get them to work, that they're open to the possibility that *_maaaaybe_* they didn't find the magical hyperparameters to make vanilla-GANs work, but that WGANs were producing reasonable results out of the gate

so they focused on WGANs. All of that sounds perfectly reasonable to me, as it's clear to everyone that the purpose of this paper isn't to benchmark GANs vs WGANs or to even give GANs a fair shot, it's to report your experience to the readers so that they can make an informed decision about what to try. Even if your GAN-vs-WGAN investigations weren't 100% systematic I think this information is valuable and I encourage the authors to mention it, especially given that the paper will likely be accepted (nobody's going to harangue you on Twitter over the GAN hyperparameters).

We agree and have included a couple of sentences on 'vanilla GANs' in the revised manuscript methods section **'On a side note, early attempts to train an original GAN on scRNA-seq data never yielded convergence, while an out-of-the-box implantation of a WGAN did. This does not imply that it is impossible to successfully train an original GAN on such data.'**

2. On describing WGAN: I think the 'precised definition' of the training objective should be moved out of the main text and into the methods section. The authors clearly disagree in their rebuttal, and that's ok, but this is still my editorial advice.

In principle we agree with the reviewer and it might be that we simply have a different definition of what constitutes the 'main text'. We considered the main text to comprise the Introduction, Results and Discussion sections, leaving the entirety of the technical description of the WGAN (e.g. the definition of the training objective) out of the main text.

We could opt for a short methods section covering the essentials and an extended supplementary information section, which would contain the more technical details such as the definition of the training objective, as suggested.

3. Dimensionality of GAN latent space: It still bugs me that the paper says that a defining characteristic of GANs is to transform a "low" dimensional representation into a "high" dimensional one. But ok, I suppose that's often the case in practice.

We are a bit puzzled by this comment and think we might be missing the reviewer's point. For clarity, what we refer to by "low dimensional representation" is the distribution over the latent space (a standard Gaussian in R^{128}), as opposed to the high-dimensional one which is an unknown, implicitly defined distribution over the gene space (roughly $R^{20,000}$). Does Reviewer 2 mean that it could be envisioned to train a GAN with a latent space that has the same apparent dimensionality as the gene space? If this is the case, it is technically true but would probably fail as it would imply implicitly mapping the cells, who most likely live on a much lower-dimensional manifold, onto a dense distribution of higher-dimension, which is a very difficult task (like trying to fill a surface by folding an infinitely thin line).

4. - If you need to cut citations, two t-SNE citations seems redundant. (Also, citation style for van der Maaten is inconsistent.)

We also corrected the inconsistency in the style. The first reference is the original one, introducing t-SNE while the second describes the Barnes-Hut approximation that we

(and pretty much everyone else) is actually using for better computational scaling of the method.

5. - Acronym WGAN is defined twice.

We removed the second definition in the revised manuscript.

[Redacted]

[Redacted]

[Redacted]

[Redacted]

[Redacted]

[Redacted]

APPENDIX A: copy of our response to Reviewer 1 in revision 3:

Reviewer #1 (Remarks to the Author):

We would like to thank reviewer 1 for the time and effort spent to this manuscript scientifically stronger. While we do not agree with reviewer 1 in all matters, the comments and criticism have not only improved this manuscript but also increased the authors' notion of 'realism' and 'partitioning'.

1. While the authors addressed many of the concerns, I still find that a major problem here is that the authors keep focusing on generating rough "clusters" of cells and not on using a sophisticated method like a GAN to generate in all the richness of the cellular phenotypic state space.

The main criticism reviewer 1 is that (point 1) 'the authors keep focusing on generating rough "clusters" of cells and not on ... to generate all the richness of the cellular phenotypic state space'. More specifically, the reviewer is not happy with selecting partitioning of cells based on clustering algorithms and would rather concentrate on (points 2 & 3 & 5) '... transitional and rare populations.' Especially for rare populations the reviewer suggests to use density estimation to sample from or condition on rare populations, which is at the heart of SUGAR (points 3 & 4) 'The whole point of the SUGAR comparison was to check how well scGAN generates transitional and rare populations'.

We understand that the reviewer has specific interest in the analysis of a continuum of cells or low density areas. The focus of this manuscript is, however, that (c)scGANs are superior to other tested algorithms in 'realistic' scRNA-seq data generation, which can then be used to augment sparse real data. The focus is not on automatically extracting which regions might be the most undersampled or 'interesting', which is the case for SUGAR, for example. In a nutshell, (c)scGANs work on arbitrary partitioning choices, including a continuum or low-density regions. To highlight this important point, we added the following sentences to the discussion of the revised manuscript: **"Throughout this manuscript, we solely focused on using cell types as a side information to condition the generation on. It is worth mentioning that any other kind of side information (partitioning of the sample) could equally be used. For instance, a cscGAN could be conditioned and trained on a combination of case and control samples. While many other choices could lead to interesting applications, we leave this avenue of research for future work."**

The choice to focus on clusters is driven by the fact that most biologists (we know) that work with scRNA-seq data care about cell types, and these occur mostly in clusters. To us it was therefore a natural choice to model and augment cell types (clusters) with a clear biological meaning. We did not and do not want to confound the clear message of the manuscript with notions of other partitions that might, and might not carry clear biological meaning.

This does not imply that (c)scGANs only work on 'rough clusters of cells'. This is especially true for scGANs for which cell generation is independent of the partitioning algorithms (e.g. density estimation, different clustering algorithms). Generated cells can be extracted *ad libitum* from partitions using any algorithm, as we have highlighted

in the answer to the 2nd major concern of reviewer 1 in the first revision. In the manuscript we had added after the first revision: “While specific cell types of interest can be obtained by scGAN cell generation followed by clustering and cell selection, we developed and evaluated various conditional scGAN architectures that can directly generate cell types of interest.”. cscGANs on the other hand are conditioned on the partitions and could theoretically be affected by the choice of algorithm. This is actually the rightful concern of reviewer 2, which we have addressed in the revised manuscript (revision 1 minor comment 7, manuscript Figure S11). It is obvious that we cannot exhaustively test all partitioning algorithms (density, cluster, etc.), which is why we did not extend this section beyond two partitioning algorithms. Again, this would only apply to the cscGAN version, the scGAN is not affected by definition.

In a nutshell, this manuscript is about realistic simulation and augmentation of scRNA-seq data. We chose to partition on cell types (clusters), which is easy to understand and a biologically valuable use case. This manuscript is complex as is and we feel if we now delve into alternate partitioning we would lose the simple message that (c)scGANs are excellent for scRNA-seq data generation and augmentation (please refer to Figure 3).

Lastly, we would like to note that (c)scGANs generate cells that cover ‘all the richness of the cellular phenotypic state space’ and we have not witnessed a single case where a ‘continuum’ or a ‘sparse area’ of cells was not modeled realistically (see Fig. 1A, 2A, 2B)).

2. Not all cell systems have highly clustered cells. More importantly, it is unclear what the value would be of artificially generating dominant cluster centroid cell types. This would be like repeatedly generating a very canonical Tcell type, perhaps naive cells over and over again. The interesting part of the data, one that is undersampled and NEEDS artificial generation may be a rare regulatory phenotype. In addition to rare populations, indeed there are many systems that are transitional, and even within a cluster there are transitions. The real utility again would be to see if it can generate through the state space.

In addition to our answer to point 1, we would like to note that a cluster has no notion of biology, it just happens to coincide with a ‘relatively clean’ population of known cell types. These cell types have been defined *a priori*, using various data types and measures. This directly implies that clusters formed by the clustering algorithm can contain arbitrary many cell types, transition states, or just one ‘clean’ cell type. Many times this is dependent not only on the algorithm but also on the parameter settings. Exactly the same holds true for any other partitioning.

In addition, we would like to highlight that (c)scGANs artificially generate cells resembling real cells in all their variability and not only to the centroids, following our definition of “realistic generation”. We would like to highlight that we have addressed this point already in the answer to the 2nd major concern of reviewer 1 in the first revision and in the previous point.

The main message of this manuscript is that (c)scGANs are superior to other tested algorithms in ‘realistic’ scRNA-seq data generation, which can then be used to augment sparse real data. The focus is not on automatically extracting which regions might be the most undersampled or ‘interesting’.

3. The whole point of the SUGAR comparison was to check how well scGAN generates transitional and rare populations. The authors repeatedly miss the point here and compare blindly. So for this you would zoom in on one cluster or continuum and see how well different parts of the data space in a continual transitional space would be generated. Are there undersampled populations that are filled in by the scGAN? What is the expected time to fill in the population? Could we conditionally generate undersampled populations? That would be really of value.

The first part of point 3 is about partitioning, which we have addressed in our answers to points 1 & 2. Here we will focus on answering the three specific questions that reviewer 1 raises.

First of all, human language is ambiguous and we are truly sorry to 'repeatedly miss the point and compare blindly'.

Q1: Are there undersampled populations that are filled in by the scGAN? (c)scGANs model all cells and all 'clusters', sparse and non-sparse ones, and we are not aware that any populations whatsoever are missed, as we stated previously. In part 4 of the results (cscGAN learns and translates gene regulatory syntax) we simulate undersampling by holding out large portions of a specific cluster from the original data and indeed show that cscGAN does fill undersampled populations (see Fig. S17).

Q2: What is the expected time to fill in the population? This would depend on the computer, on the sparsity of the cluster, and on the number of generated cells envisioned. Due to this strong conditional dependence we have taken out runtime estimates from the first version of the manuscript (revision 1, reviewer 2, point 6; manuscript section 'Realistic modeling of cells across tissues, organisms, and data size').

Generating one cell corresponds to one forward pass in a network comprising 3 fully connected layers, which is a matter of milliseconds, and the number of cells generated scale linearly in time. We have not re-included this information in the manuscript but would do so if required.

Q3: Could we conditionally generate undersampled populations? cscGANs can generate realistic cells from partitions of "undersampled" populations. The conditional generation of sparse partitions (in our case a cell cluster) is the focus of parts 2 and 4 of the manuscript. This point is also related to our answer to the 2nd comment of Reviewer 1 in the first revision ('It seems that the workaround...').

4. I also find the use of MMD to compare between sugar and scGAN to be nonsensical. As is stated in the SUGAR manuscript, SUGAR does not replicate density and EQUALIZES it. So therefore, it makes no sense to use an estimate of probability density to compare. Instead you would check if the cells generated by SUGAR and scGAN are realistic cell types, how far of outliers they are, and if rare populations are well represented.

One metric would be how well scGAN and SUGAR generate sparse parts of the dataspace (that are still well within the dataspace). You could pick these out by performing knn based density estimation on your data graph.

We agree that given our definition of “realistic generation” from which our performance measures and whole approach derives, it is only natural that an algorithm like SUGAR would fall short if it would equalize distributions. In other words, if SUGAR would sample preferentially from sparse regions in a cluster, MMD scores and also classification results compared to the full cluster distribution would indeed be ‘nonsensical’ (although we would prefer the word ‘incomparable’). To avoid confusion we would like to emphasize this problem, SUGAR with ‘density equalization’ samples preferentially from sparse areas and does therefore not reflect the true distribution of a cluster, which obviously results in higher MMD (and also classification) scores.

Density equalization: The first revision of this manuscript contained a comparison to SUGAR without ‘density equalization’ for reasons given above. In more detail, SUGAR without ‘density equalization’ samples generated cells close to every real training cell by adding ‘noise’ (estimated or fixed covariance, see below). This sampling process replicates the true distribution of the data, which would make MMD and classification a good choice to measure the realism of generated cells.

Reviewer 1 asked us in revision 2, major comment 2 to use SUGAR with ‘density equalization’ and imputation. We heeded to this specific request, which in retrospect was a mistake we corrected in this revised manuscript (see below).

Covariance estimation: While we cannot use ‘density equalization’ due to the above reasons, SUGAR’s ‘covariance estimation’ does not scale to the required number of genes and cells, as we clearly stated in revision 2 (rebuttal to Major Comment 2 of reviewer 1 in the second revision, and in the ‘SUGAR comparison’ methods section of the revised manuscript). In more detail, we argue that a generated cell can be considered realistic (up to our definition) only when it can be produced in its original space (i.e. with counts for the circa 20k genes present in the PBMC dataset, for instance). SUGAR with noise covariance estimation seems to scale quadratically in space with the number of genes and quadratically in space with the number of cells (see Figure 1 in the rebuttal to major comment 2, reviewer 1, second revision). As a result, we were not able to use SUGAR and ‘covariance estimation’ for more than 3k cells (instead of 68k to over 1 million) and 2k genes (instead of almost 20k), requiring 1.3 Tbytes of RAM and 36 hours to generate 3000 new cells from this reduced dataset (on a high-end server with 128 x 2.6GHz Xeon threads and 1.47 Tbytes of RAM). We would also like to remind reviewer 1 that the suggestion to deactivate the noise covariance estimation came directly from the first author of the SUGAR manuscript. To summarize this paragraph, SUGAR with ‘covariance estimation’ does not scale to the necessary number of genes and cells to meet our criterion of realism.

MMD: As stated above, MMD measures distribution similarity and is well suited to compare generated to real cells when SUGAR is used without ‘density equalization’. In this context we would like to thank reviewer 1 again for suggesting to use a notion of distribution (reviewer 1, revision 1, main point 3, ‘It would be better to compare results using some notion of distribution distances.’). Following this excellent

suggestion we used MMD (in addition to tSNE, correlation, and classification) to measure cell realism.

In summary, given our definition of realism and the evaluation criteria used (MMD and classification accuracy), we reverted to the results we obtained in our first revision, where we had deactivated SUGAR's noise covariance estimation and the equalization. This gave rise to the following changes in our revised manuscript: In the first part of the results section: **“SUGAR, on the other hand, generates cells that overlap with every cluster of the data it was trained on in t-SNE visualizations and accurately reflects cluster-specific gene dependencies (Fig. S6). SUGAR's MMD (59.45) and AUC (0.98), however, are significantly higher than the MMD (0.87) and AUC (0.65) of the scGAN and the MMD (0.03) and AUC (0.52) of the real data (Table S2A, Fig. S6)”**. We also adapted the 'SUGAR comparison' methods section to reflect the above line of reasoning. Finally we removed the part about the 'Reduced PBMC dataset' in the 'Datasets' section of our methods as it is no longer used.

5. Therefore I suggest in the next revision to include a section that addresses cell continuums and sparse areas of the cellular state space.

While we understand the reviewer's pique interest in a cell continuum and sparse areas of the cellular state space, we feel it might be best addressed in a separate manuscript. We fear that a section on a cell continuum and sparse areas of cellular state space would distract from the main message of realistic scRNA-seq generation and augmentation. We also don't believe a single section would suffice to address the rather complex notions of a cell continuum and sparse areas of cellular state space, both biologically and technically.

APPENDIX B : scGAN is able to learn the continuous cells states

In this Analysis, we investigate the ability and the fidelity of scGAN model (the non-conditional version) to generate scRNA-seq data corresponding to different cell states. For this purpose, we opted for the Pseudotime and PAGA-initialized topology-preserving embedding using partition-based graph abstraction.

The Pseudotime and PAGA-based analysis of single cell data assume that data lie on a connected manifold and map the inter-cell distances on a manifold. While clustering enables understanding the biological signals within cell populations, trajectory analysis using Pseudotime and graph embeddings allow the interpretation of the continuous phenotypes and processes such as development and disease progression with the continuum presented by all cells populations [Wolf et al. (2019)].

scRNA-seq data of Hematopoiesis is an ideal choice to evaluate and investigate the performance of such trajectory analysis. In essence, Hematopoiesis data [Paul et al. (2015), <http://doi.org/10.1016/j.cell.2015.11.013>] represents one of the most extensively characterized systems with stem cells differentiating through different transient cell states toward multiple cell fates.

In order to examine the ability of scGAN model to learn the manifold the data lies on, we run the PAGA and Pseudotime analysis as described in the official git repository on the Hematopoiesis scRNA-seq data [<https://github.com/theislab/paga>]. The cells were first preprocessed using the Zheng preprocessing pipeline, afterwards the force-directed single-

cell graph was built using 20 PCA components [Characterization of the single-cell transcriptional landscape by highly multiplex RNA-seq, Genome Research] [Jacomy et al. (2014), ForceAtlas2, a Continuous Graph Layout Algorithm for Handy Network Visualization Designed for the Gephi Software PLOS One.] [ForceAtlas2 for Python and NetworkX, GitHub.]. The graph is then denoised and rebuilt using PAGA-initialization as described in the official tutorial of PAGA. Figure 1 shows the graph of the Hematopoiesis scRNA-seq data. Furthermore, we run the Scanpy implementation of the Pseudotime analysis which is used to Infer progression of cells through geodesic distance along the graph [Haghverdi et al. (2016), Diffusion pseudotime robustly reconstructs branching cellular lineages, Nature Methods.].

Next step was to downsample a specific transient cell state represented by cells grouped in node 4 of the abstracted built PAGA graph (Figure 2).

The original 4th louvain group population was 150 cells, and the downsampled one is 13 cells. The downsampled scRNA-seq data (the training data) was then used to train scGAN model without providing any prior information about the cell's states or clusters. After training, the model was used to generate cells that we compared to the original dataset. After investigating the Force-directed single-cell graph of the generated cells combined with original cells we noticed that the generated cells were covering all cell states and adhering to the topology of the original scRNA-seq data.

This results presented in Figure 3 motivates us to check the fidelity of the scGAN to learn the downsampled cell state presented by the downsampled transient state 4. Therefore, we searched within the generated cells for cells that are close to the resulted sparse area which was resulted by the downsampling. A group of 137 generated cells were found and added to the downsampled scRNA-seq data. We refer to this combined group of generated and downsampled cells as augmented cells. Our assumption is that the generated cells recover the lost biological signal represented by the downsampled transient state. To prove this assumption, we plotted the Force-directed graph single-cell graph of the augmented data and compared it with the one built from the downsampled data.

The cells graph embeddings were recomputed using PAGA-initialization so that the cells are structured in a meaningful topology-preserving layout that reflects the real cell-cell interconnections and the paths of single cells. As shown in the presented Figure 4, the scGAN model (using layers with reduced numbers of neurons to accommodate for the small size of the dataset) while trained on small group of the transient states cells was still able to recover and strengthen the biological path which the held-out cells are part of.

Figure 1 : t-SNE plots of the original hematopoiesis dataset with overlay of clusterings and pseudo-time inferred by PAGA.

Figure 2 : t-SNE plots of the downsampled hematopoiesis dataset with overlay of clusterings and pseudo-time inferred by PAGA. The downsampling is affecting the structure of the graph around the clusters 0, 4, 1 and 11.

Figure 3 : t-SNE plots of real and scGAN-generated cells.

Figure 4 : t-SNE plots of the augmented hematopoiesis dataset with overlay of clusterings and pseudo-time inferred by PAGA, restoring the original structure of the graph that was lost with downsampling.

APPENDIX C: scalability of SUGAR

Fig. 5: Runtime (first column) and memory (second column) usage of SUGAR for increasing number of genes (first row) and cells (second row). SUGAR used adaptive kernel estimation

and density equalization to generate 10,000 cells. The experiments were performed using a Matlab SUGAR implementation on a Dell Power Edge R940 server with 128 x 2.6 GHz Intel Xeon threads and 1.47 Tera byte of RAM. In the end, the largest model we could fit using SUGAR was limited to 3,000 cells and 2,000 genes of the PBMC scRNA-seq dataset, using 1.3 Terabytes of RAM and taking more than 36 hours to complete.

APPENDIX D: on the stability of WGAN

Reviewer 2 in revision 1:

For the sentence "Qualitative assessment using t-SNE visualization shows that the scGAN generates cells that represent every cluster of the data it was trained on," if the authors had tried another GAN and found it wanting (in terms of mode collapse), this seems like a good place to include the result showing that WGAN hits more modes than the alternative (assuming fair attempt at tuning the alternative). Again, this would be counter to the "All GANs Created Equal?" paper, but may be true for scRNA-seq still.

While we did observe mode collapse with classical GANs on MNIST data in our preliminary work, we simply failed at managing to train them on single cell data (the iterates get very unstable and start diverging early in the training). Despite our efforts, we failed at producing anything mildly resembling any single cell sample with a classical GAN. On the other hand, the training with a WGAN required quite some tuning to get to this level of realism but even our first attempts (with the default parameters from the WGAN paper and a vanilla MLP for the critic and generator networks) yielded promising results. As one could always argue that we didn't try enough or used the wrong hyper-parameters for the classic GAN, we elected not to report those results in more detail.

APPENDIX E: on the RF classification setup and the PCA representation

Our response to Reviewer 2 in revision 1:

1. On how RF classification difficulty is evaluated:

i) The manuscript shows that a random forest struggles to classify real from simulated data when both are represented by the 50 principle components of the ~17,000 gene expression values (Cell Ranger dimensionality reduction). This is akin to showing how good a face generator is by first doing PCA on a dataset of faces and then showing that a random forest struggles to classify on that lower-dimensional representation. I agree that this result is meaningful in relative terms, i.e. PCA should capture much of what makes the data difficult to separate and that a lower AUC there is suggestive of a better generator. But, in absolute terms, it is hard to say whether a real-vs-fake classification AUC of 0.65 indicates progress in capturing the data distribution over, say, mixture models. I don't have the expertise to dictate what exact baseline is state of the art for capturing single cell RNA-seq, but I think it's appropriate include a mixture model which can easily be sampled from, such as ZINB (zero-inflated negative binomial) <https://genomebiology.biomedcentral.com/articles/10.1186/s13059-018-1406-4>, even if a tool like Splatter can't infer reasonable parameters from data (see later comments). If the authors feel strongly otherwise or think that I have misunderstood the point of demonstrating the fidelity of their generator, I will try to understand their response, but the onus is on them to clarify in the manuscript as well.

ii) Also, as a quick sanity check, at the end of training, what is the accuracy of the discriminator at classifying real versus generated data? If the AUC of the discriminator is close to 0.5 then the neural network is (surprisingly) worse than the random forest, but if the AUC is high then it suggests that a neural network would have no trouble telling the real from simulated data in its full gene-level representation.

We will start by answering sub-comment (ii) 'what is the AUC of the discriminator of the network'. This is indeed an interesting question for classical GANs, which rely on a discriminator (classifier) as a proxy to estimate the Jensen-Shannon divergence between the distributions of the real and of the generated samples. Wasserstein GANs, however, have a critic instead of a discriminator, which serves as a proxy to estimate the Wasserstein distance. They do not have a classifier. As a consequence, it is unfortunately impossible for us to assess the fidelity of the (c)scGAN using its AUC.

Now to sub-comment (i), the comparison to existing scRNA-seq simulation tools and the 'meaning' of an AUC of '0.65'. In a nutshell, and quite disappointingly, we only found a single published tool, namely SUGAR (please see comments of reviewer 1) that was able to model realistic cells. While SUGAR generates cells that give good results in t-SNE visualization and gene-gene correlation, it gives significantly higher classification and MMD scores than the scGAN. We have included this information in the updated manuscript in the main text 'Finally, we compared the results of our scGAN model to two state-of-the-art scRNA-seq simulations tools, Splatter and SUGAR (see Methods for details). While Splatter models some marginal distribution of the read counts well (Fig. S5), it struggles to learn the joint distribution of these counts, as observed in t-SNE visualizations with one homogenous cluster instead of the different subpopulations of cells of the real data, a lack of cluster-specific gene dependencies, and a high MMD score (129.52) (Table S2a, Fig. S4). SUGAR, on the other hand, generates cells that overlap with every cluster of the data it was trained on in t-SNE visualizations and accurately reflects cluster-specific gene dependencies (Fig. S6). SUGAR's MMD (59.45) and AUC (0.98), however, are significantly higher than the MMD (0.87) and AUC (0.65) of the scGAN and the MMD (0.03) and AUC (0.52) of the real data (Table S2a, Fig. S6).', Table S2, and Fig. S6.

All other tools we tested, including but not limited to ZINB-WaVE (suggested by the reviewer) and Lun2, suffer from some of the same limitations as Splatter. They model the matrix as a whole and not each gene specifically. The data produced by those methods give expression levels of "virtual genes", that have no one-to-one correspondence with real genes. While comparing t-SNE plots of cells generated by those models makes sense, using a classifier to discriminate against real cells does not since there is no virtual to real gene correspondence.

We claim, however, that the AUC obtained (where it applies) gives some measure of the realism of the cells and that they can be compared with one another to assess which model produces the more realistic cells. In this context we would like to highlight that classification is not the only quantitative measure of 'realism' we have used in the manuscript. Realistic generation of cells was also measured by gene-gene correlation and in the updated manuscript by MMD, all of which support the notion that (c)scGANs generate high fidelity cells, better than any existing tool we know of (which is additionally supported by our augmentation results).

2. On using generated samples for improving classifiers: When training the RFs (Figure 3A and Methods, "Classification of cell sub-populations").

First, using the same hyperparameters for the random forest across all subsets of data is inappropriate. One fixed set of hyperparameters will either over-fit the small samples (artificially inflating the benefit of adding samples from scGAN or cscGAN) or will under-fit the full samples. This needs to be addressed.

Second, if class weights were always used to account for class imbalance, why would duplicating some of the training points (RF upsampled) help? For example, shouldn't a weight of 2 on the smaller class build a classifier that's equivalent to duplicating all the training points in that class? Some other confusing choices in this section (70% of other cells for RF downsampled, but 100% of other cells for RF augmented).

i) We agree with the reviewer that optimizing the RF hyperparameters could increase the overall RF classification performance. While we could argue that we selected RFs and their default hyper-parameter settings because of their stability across hyperparameter space and robustness against overfitting via sample and feature bootstrapping, we do agree that hyper-parameter optimization would increase the validity of our results. We have included the results of RF hyper-parameter optimization across the number of trees and features for scRNA-seq data augmentation in the revised manuscript. In a nutshell, while the overall performance of the optimized RFs increases, RFs trained with augmented data show almost no performance loss when few real samples were available. This does not hold for upsampled or balanced approaches, replicating the initial results. We have updated the revised manuscript in the results 'In line with this assumption, the blue curves in Figure 3A and Fig. S14 show that augmenting the cluster 2 population with cluster 2 cells generated by the cscGAN almost completely mitigates the effect of the downsampling (F1 score of 0.93 obtained for a downsampling rate of 0.5%). **We obtained similar results with RFs that have been optimized for the number of trees and features per tree (Fig. S14D), showing that augmentation robustly increases classification performance across RF hyper-parameter space.**' and Fig S14.

We would like to thank the reviewer for the very helpful suggestion and believe that the hyper-parameter optimization has strengthened the results.

ii) This is a very good observation and we have to admit that we were equally puzzled when realizing the difference in performance for upsampled (RF upsampled) and downsampled (RF downsampled) training.

In a nutshell, using different class weights or upsampling are similar but not equivalent methods and yield different results. We chose these to approaches based on a literature survey of which methods might yield best results. A good overview of several approaches to deal with class imbalances is given by Chen, Liaw, and Breiman (<https://statistics.berkeley.edu/sites/default/files/tech-reports/666.pdf>).

The motivation for both experiments is that MLs, especially RFs, are very sensitive to class imbalances, resulting in models that 'ignore' small clusters.

We added a more detailed description and explanation of the 'upsampling' and 'balancing' approaches to the methods section of the revised manuscript.

- 3. On using PCA for scRNA-seq data:** Rather than PCA, consider ZIFA (zero-inflated factor analysis) which well-cited, and has a Python library available. <https://genomebiology.biomedcentral.com/articles/10.1186/s13059-015-0805-z> (Strongly recommended, though understandable if re-doing experiments and figures is too much to ask at this point. Maybe cite it as possible improvement.)

Great suggestion. In the updated manuscript we show t-SNE and classification performance of scGAN generated cells processed with ZIFA 'Of note, the fidelity with which scGANs model scRNA-seq data seems to be stable across several tested dimensionality reduction algorithms (Fig. S8).' and Fig. S8. We also cited ZIFA in the methods section of the revised manuscript. In a nutshell, ZIFA, 50 PCs, and 50 most highly-variable genes give very similar clustering, MMD, and RF classification results, providing strong evidence for the fidelity with which the scGANs learn and represent scRNA-seq data.

REVIEWERS' COMMENTS:

Reviewer #4 (Remarks to the Author):

This paper shows that, by training a generative deep neural network algorithm, the distribution of single cell RNA-seq data can be faithfully learned and further, quite realistic samples of cells generated. The authors show convincingly that their method, scGAN, can improve the quality of analysis of scRNA-seq datasets by augmenting the data, in particular small and undersampled clusters of cells. The strength of their model is that information from other, larger clusters of cells helps to learn the distribution of the undersampled clusters. The method is very much superior over SPLATTER, a method for simulation of scRNA-seq data.

The manuscript is very well written and contains a large amount of detailed backup analysis in its supplemental materials that gives confidence in the claims of the utility of the method. From a technical point of view, the study profits in a smart way from top-notch developments in machine learning. I am convinced that this method can become quite impactful in the field of scRNA transcriptomics.

Except for one central criticism, the authors have addressed all issues raised by the reviewers in a very convincing way. I will therefore only address this one critical issue here.

Both reviewers agree that the comparison of the scGAN with SUGAR by the MMD measure and the ROC measure is irrelevant and therefore misleading. They argue that, unlike csGAN, SUGAR's purpose is not to generate points according to the same probability distribution as the training data but rather to estimate the manifold in the high-dimensional data space on which the training points lie and to generate points lying on that manifold such that lowly sampled regions are more heavily sampled. Therefore, they argue, measuring SUGAR's performance by the deviation of the distributions between training data and sampled data using MMD and ROC analysis does not make sense.

The authors respond to this criticism by stating that they use SUGAR without its covariance estimation and density equalization features. In this setting, they argue, SUGAR should sample cells according to the training cell distribution. They further argue that running SUGAR in its default mode, with covariance estimation and density equalization features switched on, is infeasible on their dataset as it would require >1TB for only 3000 cells.

In my view, all these arguments, by the reviewers and authors, are valid. The reviewers are right in criticizing that SUGAR is tested on a measure that it was not designed to optimize. SUGAR and csGAN have different design goals, which makes a direct comparison problematic.

I think this dilemma can be easily solved by stressing clearly in the manuscript the caveat of the comparison in Fig. S6, namely that the comparison is somewhat theoretical as it does not test SUGAR on the task it was designed for, but that the results can be instructive nevertheless. In that context, it would be helpful for the authors to add 1-2 sentences explaining why SUGAR in its setting without equalization and covariance estimation might not sample as closely from the training distribution as scGAN.

I suggest that the authors also mention the severe runtime limitations of SUGAR in comparison to scGAN, as this is an important practical consideration for potential users. I should say that I was anyway missing an analysis of the runtime scaling of scGAN that would allow users to estimate how long scGAN would train on their own datasets, and to support their claim of sublinear scaling.

Reviewer 1 suggests to compare scGAN and SUGAR by "zooming in on one cluster or continuum

and see how well different parts of the data space in a continual transitional space would be generated. Are there undersampled populations that are filled in by the scGAN? What is the expected time to fill in the population? Could we conditionally generate undersampled populations?"

I think following this line would amount to testing scGAN on SUGAR's design goals. However, I suggest that ****at the author's discretion****, they could include their appendix B and add the SUGAR results to this analysis. It would not invalidate scGANs usefulness if SUGAR performs equally well or better at this task, as the the two tools have different goals anyway.

We would like to thank the reviewer and editor for their careful reading, comments and suggestions. We have addressed all of them, point by point, and believe that the revised manuscript is improved.

We first reply to the comments of Reviewer 4 before addressing the editorial requests.

In this rebuttal, all reviewer comments are listed in black font, while our answers are listed in blue (text changes in black).

Reviewer 4

This paper shows that, by training a generative deep neural network algorithm, the distribution of single cell RNA-seq data can be faithfully learned and further, quite realistic samples of cells generated. The authors show convincingly that their method, scGAN, can improve the quality of analysis of scRNA-seq datasets by augmenting the data, in particular small and undersampled clusters of cells. The strength of their model is that information from other, larger clusters of cells helps to learn the distribution of the undersampled clusters. The method is very much superior over SPLATTER, a method for simulation of scRNA-seq data.

The manuscript is very well written and contains a large amount of detailed backup analysis in its supplemental materials that gives confidence in the claims of the utility of the method. From a technical point of view, the study profits in a smart way from top-notch developments in machine learning. I am convinced that this method can become quite impactful in the field of scRNA transcriptomics.

Except for one central criticism, the authors have addressed all issues raised by the reviewers in a very convincing way. I will therefore only address this one critical issue here.

Both reviewers agree that the comparison of the scGAN with SUGAR by the MMD measure and the ROC measure is irrelevant and therefore misleading. They argue that, unlike csGAN, SUGAR's purpose is not to generate points according to the same probability distribution as the training data but rather to estimate the manifold in the high-dimensional data space on which the training points lie and to generate points lying on that manifold such that lowly sampled regions are more heavily sampled. Therefore, they argue, measuring SUGAR's performance by the deviation of the distributions between training data and sampled data using MMD and ROC analysis does not make sense.

The authors respond to this criticism by stating that they use SUGAR without its covariance estimation and density equalization features. In this setting, they argue, SUGAR should sample cells according to the training cell distribution. They further argue that running SUGAR in its default mode, with covariance estimation and density equalization features switched on, is infeasible on their dataset as it would require >1TB for only 3000 cells.

In my view, all these arguments, by the reviewers and authors, are valid. The reviewers are right in criticizing that SUGAR is tested on a measure that it was not designed to optimize. SUGAR and csGAN have different design goals, which makes a direct comparison problematic.

I think this dilemma can be easily solved by stressing clearly in the manuscript the caveat of the comparison in Fig. S6, namely that the comparison is somewhat theoretical as it does not test SUGAR on the task it was designed for, but that the results can be instructive nevertheless. In that context, it would be helpful for the authors to add 1-2 sentences explaining why SUGAR in its setting without equalization and covariance estimation might not sample as closely from the training distribution as scGAN.

We agree with Reviewer 4's depiction about the essence of the dilemma regarding the comparison with SUGAR, as well as with their suggestion to address it. We added the following two sentences to the corresponding section in the Results: " It is worth noting that SUGAR can be used, like here,

to generate cells that reflect the original distribution of the data. It was, however, originally designed and optimized to specifically sample cells belonging to regions of the original dataset that have a low density, which is a different task than what is covered by this manuscript." We also added the following to the corresponding section in the Methods: "It is worth noting that while the Gaussian noise, added to the real cells, is the crux of how SUGAR generates novel cells. It, however, also may be the reason why the samples produced by SUGAR do not follow the original distribution of the data as closely as those produced by scGAN."

I suggest that the authors also mention the severe runtime limitations of SUGAR in comparison to scGAN, as this is an important practical consideration for potential users. I should say that I was anyway missing an analysis of the runtime scaling of scGAN that would allow users to estimate how long scGAN would train on their own datasets, and to support their claim of sublinear scaling.

While some comments regarding the scalability of SUGAR were already present in the Methods. Following Reviewer 4's suggestion we added the following sentence in the corresponding Results section: "While SUGAR's performance might improve with the adaptive noise covariance estimation, the runtime and memory consumption for this estimation proved to be prohibitive (see Supplementary Figure 6F-I and Methods)". Finally, to substantiate those claims, we also added the panels regarding SUGAR scalability, that we had provided with our previous rebuttal, to the Supplementary Figure 6.

Regarding the analysis of the scaling of scGAN, we agree that our claims of "sublinear scaling" were not well supported. Due to the difficulty to clearly define a heuristic to adaptively stop the training beyond choosing a fixed number of epochs, we leave such study for future work. As a consequence, we modified our claim for a milder one in our Results section: "However, it should be noted that scGAN uses batch training so that its memory print does not depend on the number of cells and its runtime scales linearly, at worst, with it."

Reviewer 1 suggests to compare scGAN and SUGAR by "zooming in on one cluster or continuum and see how well different parts of the data space in a continual transitional space would be generated. Are there undersampled populations that are filled in by the scGAN? What is the expected time to fill in the population? Could we conditionally generate undersampled populations?"

I think following this line would amount to testing scGAN on SUGAR's design goals. However, I suggest that **at the author's discretion**, they could include their appendix B and add the SUGAR results to this analysis. It would not invalidate scGAN's usefulness if SUGAR performs equally well or better at this task, as the two tools have different goals anyway.

Because our appendix B contained relevant results and addressed the issue of whether our models were able to handle cells that exist in continuous rather than discrete cell states (another valid point raised by Reviewer 1), we elected to add those results in our revised manuscript. Specifically, we added the following short section in our Results: "Improved trajectory analysis using augmented data

The previous results highlight the ability of cscGAN to specifically generate cells corresponding to different types or. Such discrete states, however, are not sufficient to capture the variability of all cells within an organism.

For instance, erythrocytes are derived in the red bone marrow from pluripotent stem cells that give rise to all types of blood cells. This differentiation process can be visualized (Supplementary Figure 17A-C) using a pseudo-time analysis of bone-marrow scRNA-seq data comprising 2,730 cells²⁹ (see Supplementary Table 1 and Methods).

However, the outcome of such analyses depends heavily on how well the variety of continuous states of erythrocytes is represented in the data. To highlight this property, we manually

downsampled a sub-population of erythrocytes in the bone marrow dataset. We can observe in Supplementary Figure 17D-F that such downsampling directly affects the structure of the graph inferred by the pseudo-time analysis.

To show that scGAN can reliably model populations that exist in continuous cell states, we trained it on the downsampled bone marrow dataset. We then replaced the cells that were re-moved from the original data with handpicked cells, generated by scGAN that belonged to the same sub-population of erythrocytes. As shown in Supplementary Figure 17G-I, adding the cells generated by scGAN allows to restore the original structure of the graph. These results suggest that scGAN is able to model discrete and continuous cell states and trajectories.”. We also added the more technical detailed description (a polished version of what we had written in appendix B) to our Methods section (not reproduced it for brevity).

However, even though we agree it would have been a more suitable ground for comparison with SUGAR, the same scalability issues again prohibited it.